# Radiative pumping vs vibrational relaxation of molecular polaritons: a bosonic mapping approach

Juan B. Pérez-Sánchez [ID] & Joel Yuen-Zhou [ID] [✉]

We present a formalism to study molecular polaritons based on the bosonization of molecular vibronic states. This formalism accommodates an arbitrary number of molecules $N$, excitations and internal vibronic structures, making it ideal for investigating molecular polariton processes accounting for finite $N$ effects. We employ this formalism to rigorously derive radiative pumping and vibrational relaxation rates. We show that radiative pumping is the emission from incoherent excitons and divide its rate into transmitted and re-absorbed components. On the other hand, the vibrational relaxation rate in the weak linear vibronic coupling regime is composed of a $\mathcal{O}(1/N)$ contribution already accounted for by radiative pumping, and a $\mathcal{O}(1/N^2)$ contribution from a second-order process in the single-molecule light-matter coupling that we call polariton-assisted Raman scattering. This scattering is enhanced when the difference between fluorescence and lower polariton frequencies matches a Raman-active excitation.

Molecular exciton-polaritons are hybrid light–matter quasiparticles that emerge when the interaction strength between electronic matter excitations and confined electromagnetic fields is large enough to make Rabi oscillations faster than the molecular and cavity losses. A large variety of polaritonic architectures have been developed to reach this strong coupling regime over the last three decades, and several modifications of optical and molecular behavior have been reported[1–11]. While single molecules can strongly couple to confined fields of plasmonic nanocavities[12–16], a more common scenario requires an ensemble of matter excitations collectively coupled to optical modes in microcavities, leading to the emergence of polariton states and a dense manifold of so-called dark states[1–11]. Organic exciton-polaritons are particularly interesting systems since strong coupling between electronic and vibrational degrees of freedom (DoF) gives rise to intricate relaxation processes that allow for population transfer between dark and polariton states, a feature which plays a central role in polariton-assisted remote energy transfer[17–20], polariton transport[21–28], and polariton condensation[29–37].

Seminal works contributed to the phenomenological understanding of relaxation processes by establishing semiclassical relaxation rates valid when molecules can be treated as two-level systems weakly coupled to a vibrational bath[38–40]. Based on these works, two different mechanisms have been proposed: radiative pumping, consisting of emission from incoherent excitations directly into a polariton mode, and vibrational relaxation, where the transfer into the polariton mode is accompanied by the release of a high-frequency phonon. Despite the formal derivation of vibrational relaxation from a weak vibronic coupling model[38] and the development of theories that numerically reproduce radiative pumping[41], a rigorous derivation of vibrational relaxation and an analytical derivation of radiative pumping for molecules with complex vibrational structures within a unified framework are missing until now[27].

First-principle Hamiltonians that go beyond the Holstein–Tavis–Cummings (HTC) model have been put forward with the aim of understanding polariton-modified chemical reactivity[42,43], and recent theoretical works suggest that relaxation from polaritons to dark states in the $N \to \infty$ limit can be understood simply as an optical filtering effect: polaritons act as windows through which vibronic states can be optically excited[44,45]. This is consistent with several theoretical[46–48] and experimental[49–52] works. This striking finding has made understanding molecular polariton dynamics in the finite $N$ limit more crucial than ever for achieving nontrivial polaritonic effects in

Department of Chemistry, University of California San Diego, La Jolla, CA, USA. [✉]e-mail: joelyuen@ucsd.edu

the collective strong coupling regime, particularly the relaxation from dark states back to polariton states.

In this work, we re-derive an exact bosonic picture of organic molecular polaritons from first principles to study molecular dynamics under collective light–matter coupling for arbitrary number of molecules $N$, excitations $N_{exc}$, and internal vibrational structure of the molecules[53-60]. This mapping is analogous to the Schwinger boson representation of spins[61], and is ideal for numerical simulations using Meyer-Miller mappings[62-65], quantum cumulant expansions[66], quantum computing with bosonic devices[67], and the multi-configurational time-dependent Hartree for bosons (MCTDHB)[68,69]. Next, we focus on the large (yet finite) $N$ case and the first excitation manifold to rigorously derive radiative pumping and vibrational relaxation rates. We achieve this by partitioning the bosonic Hamiltonian into fast and slow components, treating the slow components perturbatively. Next, we unambiguously establish the fundamental differences between these two polariton relaxation mechanisms. Radiative pumping is the emission from incoherent excitons that populate the polaritons, which can either leak out of the cavity or be reabsorbed by the material. The latter, which we call polariton-assisted photon recycling, is a strong coupling phenomenon responsible for long-range energy transfer and may play an important role in polariton transport and polariton condensation. We also provide simple analytical formulas for radiative pumping and polariton-assisted photon recycling in terms of linear optical properties, which can be easily used by experimentalists without the need of quantum chemistry calculations. On the other hand, vibrational relaxation includes radiative pumping and higher-order processes in the single-molecule coupling $g$. We characterize one of such processes and call it polariton-assisted Raman scattering, which we believe corresponds to vibrationally assisted scattering (VAS)[70]. We lay down approximations to calculate polariton-assisted Raman scattering rates, which we apply on a simple model. In Fig. 1, we summarize the molecular polariton photophysics discussed in this manuscript.

## Results

### Molecular Polariton Hamiltonian

Consider a system of $N$ noninteracting molecules collectively coupled to a single-cavity mode. The Tavis–Cummings Hamiltonian, extended to include internal vibrational degrees of freedom missing from original models, can be written as (hereafter $\hbar = 1$)

$$\hat{H} = \sum_{i}^{N} \left( \hat{H}_m^{(i)} + \hat{H}_I^{(i)} \right) + \hat{H}_{cav}, \tag{1}$$

where

$$\begin{aligned} \hat{H}_{mol}^{(i)} &= \left( \hat{T} + V_g(q_i) \right)|g_i\rangle\langle g_i| + \left( \hat{T} + V_e(q_i) \right)|e_i\rangle\langle e_i|, \\ \hat{H}_{cav} &= \omega_c \hat{a}^\dagger \hat{a}, \qquad \hat{H}_I^{(i)} = g\left( |e_i\rangle\langle g_i|\hat{a} + |g_i\rangle\langle e_i|\hat{a}^\dagger \right), \end{aligned} \tag{2}$$

are the Hamiltonians for the $i$th molecule, the cavity mode, and the interaction between them, respectively. Here, $\hat{T}$ is the kinetic energy operator, $|g_i\rangle$ and $|e_i\rangle$ are the molecular ground and excited electronic states, $V_{g/e}(q_i)$ are the ground and excited Potential Energy Surfaces (PES), $\hat{a}$ is the photon annihilation operator, and $q_i$ is the vector of all vibrational degrees of freedom of molecule $i$. Here we consider only two electronic states per molecule and use the rotating wave, the Born–Oppenheimer, and the Condon approximations for convenience.

### Bosonic mapping

In our previous work we have derived Collective Dynamics Using Truncated-Equations (CUT-E), a formalism that, by exploiting the permutational symmetries of the exact time-dependent many-body

**a** Radiative pumping

**b** Polariton-assisted Raman scattering

**Fig. 1 | Schematic representation of the polariton relaxation mechanisms categorized in this work. a** Radiative pumping: a first-order process in the single-molecule light–matter coupling. An emitted photon from an incoherent exciton can either leak out of the cavity or be reabsorbed by the molecules, enabling polariton-assisted photon recycling. **b** Polariton-assisted Raman scattering: a second-order process in the single-molecule light–matter coupling. It involves the virtual emission of a photon from an incoherent exciton, followed by Raman scattering by a second molecule. The resulting red-shifted photon then leaks out of the cavity. Vibrational relaxation encompasses radiative pumping, polariton-assisted Raman scattering, and higher-order processes.

(many-molecule and cavity) wavefunction, yields a hierarchy of time-scales that renders the simulation efficient for large $N$[71]. Here, we recognize that this formalism can be easily derived my moving to a picture where molecules are treated as bosonic particles with internal structure. The bosonic mapping of identical-noninteracting particles is well-known[61–63], and it has been applied to ensembles of $d$-level systems strongly interacting with light[53–60] (also see ref. [72] for a fermionic mapping). The mapping assumes that we start from a permutationally symmetric wavefunction at an initial time. Since the Hamiltonian in Eq. (1) is permutationally symmetric, this symmetry is preserved for all times, thus allowing us to focus only on the bosonic (permutationally symmetric) subspace. We carry out this mapping by transforming the molecular operators $\hat{O} = \sum_i^N \hat{o}_i$ according to the standard recipe,

$$\hat{O} \rightarrow \sum_{i,j} \langle i|\hat{o}|j\rangle \hat{\mathcal{B}}_i^\dagger \hat{\mathcal{B}}_j, \qquad (3)$$

where $\hat{o}$ is a single-molecule operator, and $\hat{\mathcal{B}}_i$ are bosonic operators that annihilate a molecule (not an exciton) in a vibronic state $|i\rangle$. For convenience, we use the vibrational eigenstates of the ground electronic state, $|\varphi_i^{(g)}\rangle$, as a basis for the excited electronic state. This yields (see Supplementary Section 1 for a step-by-step derivation of the bosonic Hamiltonian)

$$\hat{H} = \omega_c \hat{a}^\dagger \hat{a} + \sum_i^m \omega_{g,i} \hat{b}_i^\dagger \hat{b}_i + \sum_i^m \omega_{e,i} \hat{B}_i^\dagger \hat{B}_i + \sum_{i \neq j}^m \langle \varphi_i^{(g)}|\hat{V}_{eg}|\varphi_j^{(g)}\rangle \hat{B}_i^\dagger \hat{B}_j$$
$$+ g \sum_i^m \left( \hat{B}_i^\dagger \hat{b}_i \hat{a} + \hat{B}_i \hat{b}_i^\dagger \hat{a}^\dagger \right), \qquad (4)$$

where $\hat{a}$, $\hat{b}_i$, and $\hat{B}_i$ annihilate a photon, a molecule in the vibronic state $|g, \varphi_i^{(g)}\rangle$, and a molecule in the vibronic state $|e, \varphi_i^{(g)}\rangle$, respectively. Moreover, $\hat{V}_{eg} = \hat{V}_e - \hat{V}_g$ is the total vibronic coupling operator responsible for the molecular dynamics of electronically excited molecules, $\omega_{e,i} = \langle \varphi_i^{(g)}|\hat{T} + \hat{V}_e|\varphi_i^{(g)}\rangle$, and $m$ is the size of the vibrational basis. The corresponding many-body basis states, $|n_1 n_2 \cdots n_m, n_1' n_2' \cdots n_m', N_{ph}\rangle$, are eigenstates of the noninteracting Hamiltonian (i.e., when $V_{eg,ij} = 0$ and $g = 0$), with $\sum_i^m n_i = N_g$ and $\sum_i^m n_i' = N_e$, and with $N_g$ and $N_e$ being the number of ground and excited molecules, respectively.

In this framework, absorption is seen as the destruction of a photon and a molecule in the initial vibronic state, to create a molecule in an excited vibronic state; each vibronic state corresponds to a harmonic oscillator carrying a number of excitations equal to the number of molecules in such state (see Fig. 2). This bosonization is exact for any values of $N$ and $N_{exc}$, and can be easily applied beyond the approximations to the molecular Hamiltonian mentioned above. Finally, it is easy to check that the number of excitations and the number of molecules are conserved, since $[\hat{N}_{exc}, \hat{H}] = [\hat{N}, \hat{H}] = 0$, for $\hat{N}_{exc} = \hat{a}^\dagger \hat{a} + \sum_i^m \hat{B}_i^\dagger \hat{B}_i$ and $\hat{N} = \sum_i^m \hat{b}_i^\dagger \hat{b}_i + \sum_i^m \hat{B}_i^\dagger \hat{B}_i$.

We now partition the bosonic Hamiltonian as $\hat{H} = \hat{H}_0 + \hat{H}_{vc} + \hat{H}_{sm}$, where $\hat{H}_{vc}$ contains all weak contributions to the vibronic coupling (e.g., spin–orbit couplings), while $\hat{H}_{sm}$ contains all single-molecule light–matter coupling terms $\sim g$. This partitioning allows us to define vibrational relaxation and radiative pumping rates using first-order perturbation theory on $\hat{H}_{vc}$ and $\hat{H}_{sm}$, respectively (see "Methods").

### Radiative pumping rate

We can write the eigenstates of $\hat{H}_0 + \hat{H}_{vc}$ in the first excitation manifold as (see Supplementary Section 3)

$$(\hat{H}_0 + \hat{H}_{vc})|\xi, \{n_j\}\rangle = \omega_{\xi, \{n_j\}}|\xi, \{n_j\}\rangle,$$
$$|\xi, \{n_j\}\rangle = a_{\{n_j\}}^{(\xi)}|n_1 n_2 \cdots n_m, 00 \cdots 0, 1\rangle$$
$$+ \sum_i^m b_{\{n_j\}}^{(\xi, i)}|(n_1 - 1)n_2 \cdots n_m, \cdots 1_i \cdots, 0\rangle \qquad (5)$$

where $|\xi\rangle$ are polaritons and dark vibronic states, $\{n_j\}$ are the number of molecules on each vibrational state of the electronic ground-state, $a_{\{n_j\}}^{(\xi)}$ are photonic Hopfield coefficients, and $b_{\{n_j\}}^{(\xi, i)}$ are the matter Hopfield coefficients. For simplification, we will use Philpott's notation to represent the bosonic basis states from now on (see "Methods")[73]. We can define an initial dark state that corresponds to one excited molecule in a fully Stokes-shifted configuration with negligible overlap with the FC state (the lowest vibrational state of the molecular excited PES, see Fig. 3),

$$(\hat{H}_0 + \hat{H}_{vc})|ss\rangle \approx \omega_{ss}|ss\rangle, \quad |ss\rangle = \sum_{i>1}^m c_{exc}^{(i)}|e_i\rangle. \qquad (6)$$

This dark state is an incoherent exciton that can couple to the cavity mode via single-molecule light–matter coupling $\hat{H}_{sm}$ (despite the oxymoron of an emissive dark state, we will continue using this terminology since it is widespread in the molecular polaritonics literature), and therefore differs from the dark states of the TC model whose couplings to the cavity mode vanish exactly[74–77] (see Supplementary Section 4 for a detailed comparison).

The FGR rate from $|ss\rangle$ to all possible final states $|\xi, \{n_j\}\rangle$ yields (see Supplementary Section 5 for details)

$$\Gamma_{rp} = 2\pi g^2 \sum_\xi \sum_{j>1}^m |a_{1j}^{(\xi)}|^2 |c_{exc}^{(j)}|^2 \frac{\gamma_\xi / \pi}{(\omega_\xi - (\omega_{ss} - \omega_{g,j}))^2 + \gamma_\xi^2}, \qquad (7)$$

where we renamed $a_{(N-1)\cdots 1_j \cdots}^{(\xi)} \equiv a_{1j}^{(\xi)}$, set $\omega_{g,1} = 0$, and approximated $\omega_{\xi, (N-1)\cdots 1_j \cdots} \approx \omega_\xi + \omega_{g,j}$, with $\omega_\xi = \omega_{\xi, (N-1)\cdots 0\cdots}$ being the polariton frequency and $\omega_{g,j}$ being the frequency of the phonon created during the emission. In the derivation, we have assumed $N \gg 1$ and summed over

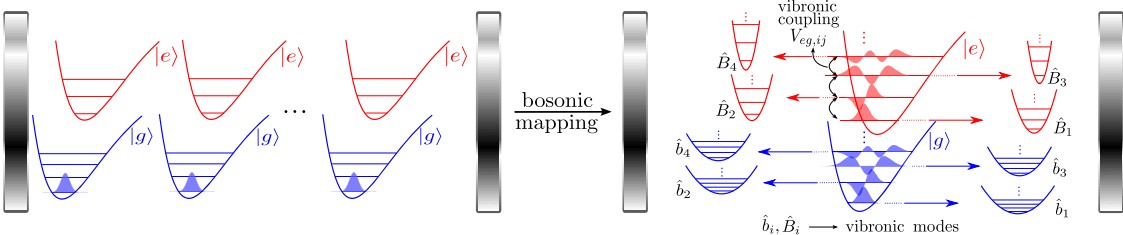

**Fig. 2 | Bosonic mapping of molecular polaritons.** A permutationally symmetric wavefunction of the entire molecular ensemble and cavity remains in the permutationally symmetric subspace throughout its evolution generated by $\hat{H}$. Hence, a dramatic simplification of the simulation can be afforded by working in the bosonic subspace. In this bosonic representation, the vibronic states $|g, \varphi_i^{(g)}\rangle$ and $|e, \varphi_i^{(g)}\rangle$ are mapped to the harmonic oscillators $\hat{b}_i$ and $\hat{B}_i$ carrying a number of excitations equal to the number of molecules in such states. Similarly, the vibronic coupling $V_{eg,ij} = \langle \varphi_i^{(g)}|\hat{V}_{eg}|\varphi_j^{(g)}\rangle$ between excited vibronic states now couples vibronic modes.

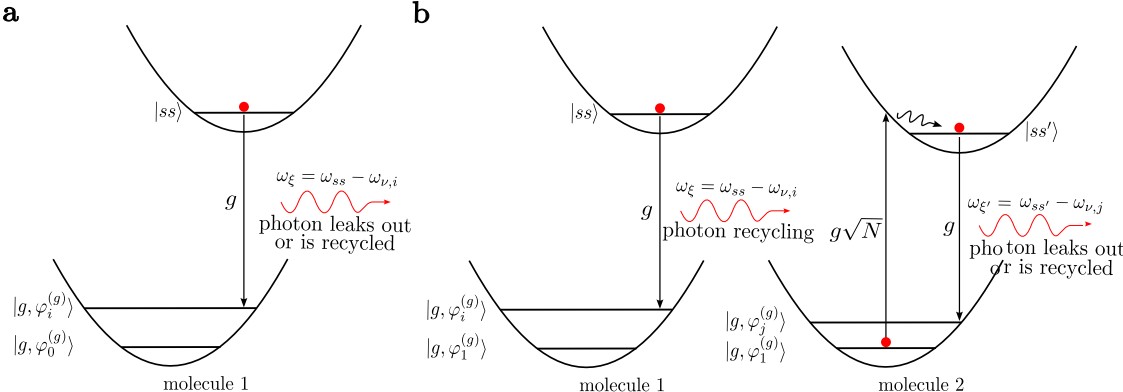

**Fig. 3 | Radiative pumping and polariton-assisted photon recycling mechanisms. a** Radiative pumping: emission from a Stokes-shifted molecule $|ss\rangle$ through a polariton state $|\xi\rangle$ (typically the LP). **b** Polariton-assisted photon recycling: light emitted from dark states is reabsorbed before it leaks out of the cavity via collective strong light–matter coupling ($g\sqrt{N} > \kappa$). This excitation creates a new Stokes-shifted molecule $|ss'\rangle$ that can subsequently re-emit. This occurs if the bare emission and absorption spectra of the material overlap.

all final eigenstates of $\hat{H}_0 + \hat{H}_{vc}$, which have a finite resolution $\gamma_\xi$ due to finite cavity lifetime $\kappa$ [molecular dissipation is not needed because every molecular interaction is in principle accounted for by Eq. (1)]. Finally, the coefficients $|c_{exc}^{(j)}|^2$ are FC factors associated with the bare molecular emission profile[78]

$$\sigma_{em}^{(out)}(\omega) = \sum_{j>1}^{m} |c_{exc}^{(j)}|^2 \delta(\omega - (\omega_{ss} - \omega_{g,j})), \quad (8)$$

which assumes that the same Stokes-shifted state is reached inside and outside the cavity[74–77].

From this analysis, it is clear that the radiative pumping rate in Eq. (7) is proportional to the fluorescence of a bare molecule at frequencies $\omega_{ss} - \omega_{g,j}$ into all final states $|\xi\rangle$, weighted by their Hopfield coefficient. This is consistent with phenomenological and experimental works[17,38,39,75,79].

### Radiative pumping as an overlap between spectroscopic observables

A more useful and insightful expression for $\Gamma_{rp}$ can be obtained by writing Eq. (7) in terms of the polaritonic linear spectroscopic observables in refs. [46,48,80] (see Supplementary Section 3.1):

$$
\begin{aligned}
\Gamma_{rp} &= \int d\omega \Gamma_{rp}(\omega) = -2g^2 \int d\omega \sigma_{em}(\omega) \text{Im}\left[D^R(\omega)\right] \\
&= \frac{2g^2}{\kappa} \int d\omega \sigma_{em}(\omega)[A(\omega) + 2T(\omega)],
\end{aligned} \quad (9)
$$

where $D^{(R)}(\omega)$, $A(\omega)$, and $T(\omega)$ are the photon–photon correlation function, the polariton absorption spectrum, and the polariton transmission spectrum, in the $N \to \infty$ limit, respectively[46,48,80]. Notice that evaluating Eq. (9) does not require expensive quantum dynamics simulations or quantum chemistry calculations, as all the information about the excited state molecular dynamics is encoded in the polariton linear response formulas, which are routinely measured. More rigorously, $D^{(R)}(\omega)$, $A(\omega)$, and $T(\omega)$ are the linear optical quantities where one of the molecules is vibrationally excited while the remaining $N - 1$ are in the global ground-state (molecules emit into polaritons with a slightly reduced Rabi splitting $g\sqrt{N-1}$, an effect negligible in the large $N$ limit). The prefactor $g^2/\kappa \propto \mathcal{Q}/\mathcal{V}_c$ encodes cavity-enhancement of the molecular emission, with $\mathcal{Q}$ being the cavity-quality factor and $\mathcal{V}_c$ being the cavity mode volume. Since $\text{Im}[D^R(\omega)]$ measures the dissipation rate of the emitted photon into the molecular and electromagnetic baths, this linear optics description provides the rate of radiative decay of $|ss\rangle$ as the sum of transmitted and reabsorbed components. In other words, it accounts for the rate from $|ss\rangle$ to polariton states and also the subsequent rates from polaritons back to dark states ($\propto A(\omega)$) or out of the cavity ($\propto T(\omega)$) (see Fig. 3).

A simple extension of Eq. (9) to the multimode scenario can be justified owing to a crucial observation. As Engelhardt and Cao recently showed, in the absence of vibronic coupling, coupling between different cavity $k_\parallel$ modes vanishes in the limit where $N \to \infty$[81]. If we assume this to hold in the presence of homogeneous broadening, this implies that $k_\parallel$ is a good quantum number for the polaritonic eigenstates of $\hat{H}_0 + \hat{H}_{vc}$, which is the total Hamiltonian in the $N \to \infty$ limit[44]. This allows us to define linear optical properties $D_{k_\parallel}^{(R)}(\omega)$, $A_{k_\parallel}(\omega)$, and $T_{k_\parallel}(\omega)$. For finite $N$, the single-molecule coupling $\hat{H}_{sm}$ that gives rise to radiative pumping couples different cavity modes. Therefore, we can estimate the radiative pumping rate in the multimode case to take the form

$$\Gamma_{rp} = \frac{2g^2}{\kappa} \sum_{k_\parallel} \int d\omega \sigma_{em}(\omega)\left[A_{k_\parallel}(\omega) + 2T_{k_\parallel}(\omega)\right], \quad (10)$$

where we have ignored any $k_\parallel$-dependence of $g$ and $\kappa$. This expression is consistent with the fact that radiative pumping is simply fluorescence from incoherent excitons, and can pump any polariton mode. However, this argument neglects Fröhlich interactions that could result from collective phonons. This limitation arises from the starting point of our model whereby no intermolecular couplings are present in the absence of the photon mode; these interactions might be more relevant in structured solids[82].

### Polariton-assisted photon recycling

Using Eq. (34) from ref. [48] relating the linear spectroscopy of polaritons to the molecular susceptibilities, we can obtain an expression for the ratio of the light emitted by dark states that is reabsorbed and transmitted[44],

$$\frac{A(\omega)}{2T(\omega)} = \mathcal{Q}\text{Im}\left[\chi^{(1)}(\omega)\right], \quad (11)$$

where $\chi^{(1)}(\omega)$ is the bare absorption spectrum of the ensemble ($\propto Ng^2$). This demonstrates that the light emitted by incoherent excitons, into the polariton states, can be reabsorbed by other molecules inside the cavity, a phenomenon which is enhanced by the collective coupling and the cavity-quality factor (see Fig. 3b). This process has recently been characterized as polariton-assisted photon recycling[83], and has also been discussed in several previous works[17–20,28,84]. It can significantly impact the photoluminescence of polaritonic systems due to re-emission of the absorbed light.

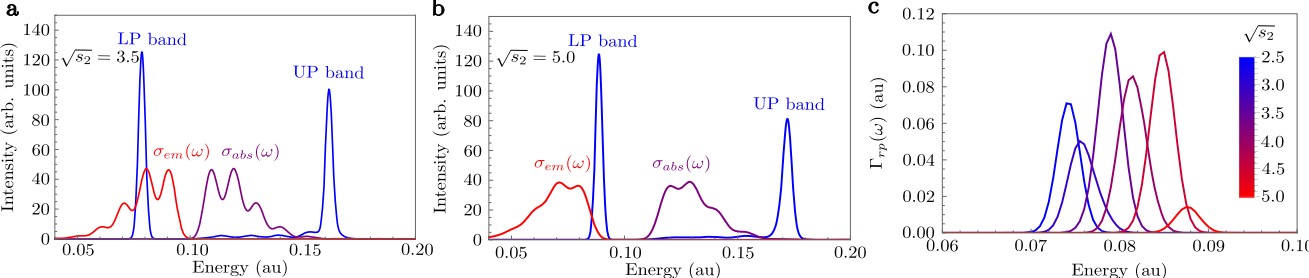

**Fig. 4 | Radiative pumping for different values of Stokes shift (∝ $s_2$). a, b** Polariton bands from the polariton absorption and transmission spectra $A(\omega) + 2T(\omega)$, bare molecular absorption profile $\sigma_{abs}(\omega)$, and bare molecular emission profile $\sigma_{em}(\omega)$, for $\sqrt{s_2} = 3.5$ (**a**) and $\sqrt{s_2} = 5.0$ (**b**). **c** The frequency-resolved radiative

pumping $\Gamma_{rp}(\omega)$ is proportional to the overlap between the polariton bands and the bare molecular emission. The total radiative pumping rate $\Gamma_{rp}$ is the integral of $\Gamma_{rp}(\omega)$.

Using Eqs. (9) and (11), the polariton-assisted photon recycling (or energy transfer) rate $\Gamma_{rec}$ is given by

$$\Gamma_{rec} = \frac{4g^2 \mathcal{Q}}{\kappa} \int d\omega \sigma_{em}(\omega) T(\omega) \text{Im}\left[\chi^{(1)}(\omega)\right]. \quad (12)$$

Although we derived this rate in the first excitation manifold, we expect it to be enhanced via bosonic stimulation if many polaritons are present in the mode that mediates the energy transfer process. Despite this effect being analogous to stimulated emission outside of the cavity, there is a unique opportunity in the strong coupling regime, which is to use it to enhance long-range energy transfer to other molecules of the ensemble. Finally, the competition between polariton-assisted photon recycling and Förster resonance energy transfer (FRET, here neglected)[19], and its dependence with the number of polaritons, is an interesting focus of study for future works.

In Fig. 4, we perform numerical simulations of radiative pumping rates for a chromophore described by a two-mode linear vibronic coupling Hamiltonian (see "Methods"). We show how radiative pumping increases when Stokes shift causes a significant overlap between the bare molecular emission and the lower polariton band. Due to the large Stokes shift, the lower polariton branch does not significantly overlap with the bare molecular absorption. As we explained above, this implies that most of the light emitted from dark states is transmitted out of the cavity.

**Vibrational relaxation in the weak vibronic coupling limit**
We obtain the vibrational relaxation rate by considering $\hat{H}_{vc}$ as a perturbation of magnitude $W \ll g$ that causes transitions between eigenstates of $\hat{H}_0 + \hat{H}_{sm}$. This treatment includes all-order processes in $\hat{H}_{sm}$. This is a natural description when collective light–matter coupling is reached with a few tens of molecules or less, or when the molecular process of interest is much slower than radiative decay (e.g., reverse intersystem crossing in organic molecules[85,86]). We show that this strategy generalizes the vibrational relaxation rate in the linear vibronic coupling limit originally studied by Litinskaya et al.[75], and allows a direct comparison between vibrational relaxation and radiative pumping rates.

The Hamiltonian $\hat{H}_0 + \hat{H}_{sm}$ can be diagonalized exactly to obtain polaritons and dark states (see Supplementary Section 5). We calculate the FGR rate from a dark initial eigenstate with a single phonon in the vibrational state $k$

$$|D_k\rangle = \sqrt{\frac{N-1}{N}}|e_k\rangle - \sqrt{\frac{1}{N}}|g_k e_1\rangle. \quad (13)$$

Notice that this dark state differs from the incoherent excitons considered by Litinskaya and coworkers (see Eq. (6))[75], mainly in the $1/\sqrt{N}$

correction that arises due to the single-molecule coupling $g$. We will show that this $1/\sqrt{N}$ correction is crucial since it gives rise to a nontrivial Raman scattering process[60] that contributes to the vibrational relaxation rate, and is not taken into account in previous works[75,87].

Assuming no detuning between cavity and exciton frequencies, the vibrational relaxation rate yields

$$\Gamma_{\xi_{\pm} \leftarrow D_k} = 2\pi\left(\frac{N-1}{2N^2}\right)\sum_{i>1\neq k}^{m}|V_{eg,ik}|^2 \frac{\gamma_{\xi_{\pm}}/\pi}{\left(\omega_{g,i}-\omega_{g,k}\pm g\sqrt{N}\right)^2 + \gamma_{\xi_{\pm}}^2}$$
$$+ 2\pi\left(\frac{1}{2N^2}\right)\sum_{i>1\neq k}^{m}|V_{eg,i1}|^2 \frac{\gamma_{\xi_{\pm}}/\pi}{\left(\omega_{g,i}-\omega_{g,1}\pm g\sqrt{N}\right)^2 + \gamma_{\xi_{\pm}}^2}. \quad (14)$$

Here, we have ignored terms that correspond to couplings from Stokes-shifted configurations directly into the FC region (excited state vibrational recurrences, which are unlikely after Stokes shift has ensued given that they involve the recoherence of a large number of vibrational modes back into the FC region), and processes in which a phonon is produced in the same vibrational mode $k$ as the initial dark state. This is a good approximation in the vibrational bath limit (see Supplementary Section 5). Equation (14) reduces to the well-known vibrational relaxation rate by Litinskaya[38,75,87] upon consideration of four additional assumptions. Two of them are: (a) removal of the second term (not really justified) and (b) in the linear vibronic coupling limit. This can be easily seen by noticing that the sum over vibronic states $i > 1$ can be changed for a sum over vibrational modes, given that each mode of the vibrational bath will contribute with a single state:

$$\Gamma_{\xi_{\pm}\leftarrow D_k} \approx 2\pi\left(\frac{N-1}{2N^2}\right)\sum_i \omega_{\nu,i}^2 s_i \frac{\gamma_{\xi_{\pm}}/\pi}{\left(\omega_{\nu,i} \pm g\sqrt{N}\right)^2 + \gamma_{\xi_{\pm}}^2}. \quad (15)$$

The other two additional assumptions are: (c) $(N-1)/N^2 \approx 1/N$ when $N \gg 1$, (d) there are many photon modes in the cavity (given the single-mode assumption in our derivation, we do not attempt further comparison). Regardless, the main physics we are interested in is the second term in Eq. (14) which has been missing in the literature throughout.

**Vibrational relaxation vs radiative pumping**
We now interpret the mechanisms involved in the vibrational relaxation rate. We do so by looking at the initial and final states in the FGR rate that generate each of the two terms in Eq. (14) (see Supplementary Section 6 for a more detailed analysis). We find that the first term ($\propto \frac{N-1}{N^2}$) is a first-order process in $g$, and can be interpreted as single phonon emission from an incoherent exciton followed by emission,

i.e., the tails of the emission spectra: $|e_k\rangle \xrightarrow{W} |e_i\rangle \xrightarrow{g} |g_i1\rangle$. Notice that this mechanism is already included in the radiative pumping rate (see Fig. 3). On the other hand, the second term of Eq. (14) ($\propto \frac{1}{N^2}$) is a second-order process in $g$ that consists of the virtual emission from an incoherent exciton, followed by polariton-assisted Raman scattering into a lower energy polariton (see Fig. 5): $|e_k\rangle \xrightarrow{g} |g_i1\rangle \xrightarrow{g\sqrt{N-1}} |g_k e_1\rangle \xrightarrow{W}$ $|g_k e_i\rangle \xrightarrow{g} |g_k g_i1\rangle$; the frequency of the actual photon emission is equal to the emission frequency of the first molecule minus the energy of the phonons created in the second molecule. Notice that this is just the coherent version of polariton-assisted photon recycling in Fig. 3b. A more interesting scenario arises when strong vibronic coupling is present, which can play the role of the weak vibronic coupling $W$ in the above mechanism. This can enhance Raman scattering and allow vibrational relaxation to include third and higher-order processes in $g$ (see Supplementary Section 7, and Supplementary Fig. 1). However, based on the analysis detailed in Supplementary Section 7 and our previous work[71], each of these scattering processes would be penalized by a $1/N$ factor in the rate, rendering those processes relevant only for small $N$, long-time dynamics, or large number of excitations.

Based on the analysis above, we can characterize each mechanism that contributes to the vibrational relaxation rate by taking the single-molecule coupling $g$ as the perturbation, granted that all orders of perturbation theory are considered (fluorescence, Raman, hyper-Raman, etc.[88]). Since this approach leads to a much more intuitive understanding of the mechanisms involve in the

relaxation between dark and polaritons states, we advise the community to stick to the radiative pumping framework. In the next section, we derive the rate for the polariton-assisted Raman scattering mechanism of Fig. 5, in the strong vibronic coupling regime, using the aforementioned approach.

### Polariton-assisted Raman scattering

We calculate the polariton-assisted Raman scattering rate using second-order perturbation theory on the single-molecule light–matter coupling $\hat{H}_{sm}$. As for radiative pumping, the Stokes-shifted state $|ss\rangle$ is chosen as the initial dark state. This calculation involves summing over all final states that have two additional ground-state molecules with phonons (the first molecule acquires phonons via virtual emission, and the second molecule acquires phonons via Raman scattering),

$$\Gamma_{scatt} = 2\pi \sum_{\xi',\{n'_j\}} |A_{\xi',\{n'_j\}\leftarrow ss}|^2 \frac{\gamma_{\xi'}/\pi}{\left(\omega_{\xi',\{n'_j\}} - \omega_{ss}\right)^2 + \gamma_{\xi'}^2}$$

$$A_{\xi',\{n'_j\}\leftarrow ss} = \sum_{\xi,\{n_j\}} \frac{\langle\xi',\{n'_j\}|\hat{H}_{sm}|\xi,\{n_j\}\rangle\langle\xi,\{n_j\}|\hat{H}_{sm}|ss\rangle}{\omega_{\xi,\{n_j\}} - \omega_{ss} + i\gamma_{\xi}}.$$

(16)

The scattering rate is fairly easy to calculate in the large $N$ (yet finite) limit, even in the presence of strong vibronic coupling (see details in Supplementary Section 8). In Fig. 6, we compute $\Gamma_{scatt}$ for the model system introduced in Eq. (19) for different values of Stokes shift.

Besides the phonons released during the virtual emission $\omega_{\nu,i}$ from the incoherent dark state $|ss\rangle$, the scattering mechanism also creates phonons via Raman scattering on a second molecule $\omega_{\nu,j}$. As a consequence, the scattering relaxation rate does not rely on the overlap between the bare emission and the lower polariton band. Instead, it requires that the difference between the emission and the lower polariton $|\xi'\rangle$ is compensated by the vibrational excitation created in the Raman process, i.e., $\omega_{\xi'} = \omega_{ss} - \omega_{\nu_i} - \omega_{\nu_j}$ (note that $i$ and $j$ are vibrational states that can represent single or multiple phonons). We illustrate this phenomenon by shifting the lower polariton by $\omega_{\nu,1}$ in Fig. 6b (blue-dashed band), although all vibrational states coupled to the FC region (including those with more than one phonon) contribute to the rate. Importantly, $\omega_{\nu_i}$ also produces vibronic progressions in the bare molecular emission spectrum. This means that resonant conditions for radiative pumping and Raman scattering always coexist, making it challenging to identify which relaxation mechanism is in play. This explains why Tichauer and coworkers found that vibrational relaxation and radiative pumping are driven by similar vibrational modes[89].

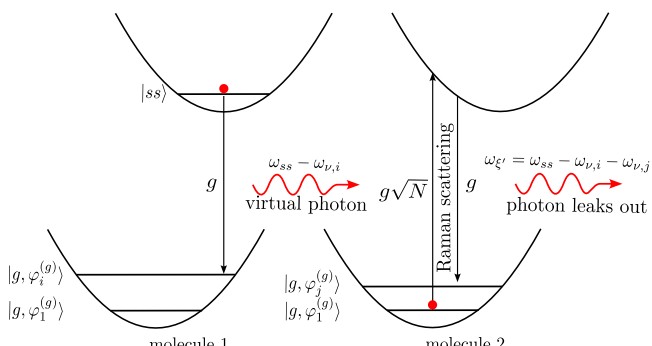

**Fig. 5 | Polariton-assisted Raman scattering mechanism.** Second-order process in the single-molecule light–matter coupling $g$ that results in the release of a photon, and vibrational excitations $\omega_{\nu_i}$ and $\omega_{\nu_j}$ in two different molecules (even when they are far apart). It is fundamentally different to radiative pumping and can be regarded as a coherent version of the polariton-assisted photon recycling mechanism in Fig. 3b.

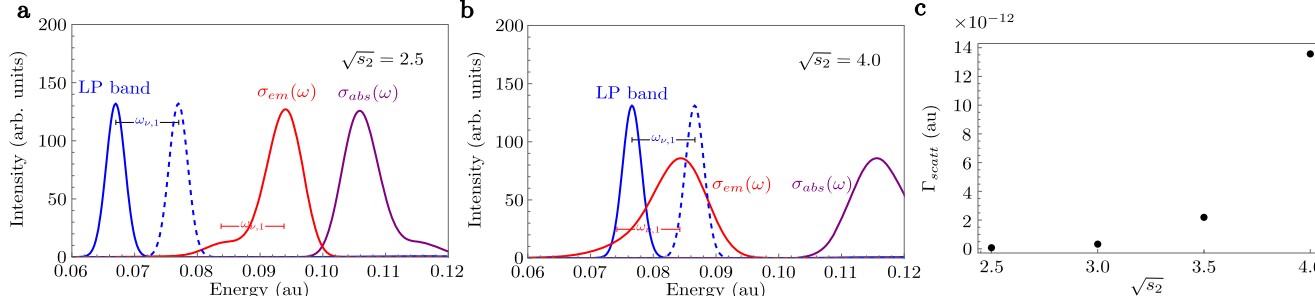

**Fig. 6 | Polariton Raman scattering rate for different values of Stokes shift** ($\propto s_2$). **a, b** Polariton bands from the polariton transmission spectra $T(\omega)$, bare molecular absorption profile $\sigma_{abs}(\omega)$, and bare molecular emission profile $\sigma_{em}(\omega)$, for $\sqrt{s_2} = 2.5$ (**a**) and $\sqrt{s_2} = 4.0$ (**b**). **c** Decay rate from dark to lower polariton via Raman scattering $\Gamma_{scatt}$. The rate increases when the energy difference between the

emission and the lower polariton corresponds to the vibrational excitation created in the Raman process, e.g., $\omega_{\nu,1}$. Yet, $\Gamma_{scatt}$ is quite low since the broadening $\gamma$ is quite large, there are no polaritons to resonantly scatter from due to the single-mode nature of the model, and we do not include non-Condon effects.

Finally, we believe polariton-assisted Raman scattering corresponds to VAS[70,89–92]. Although our calculations in Fig. 6c show that $\Gamma_{scatt}$ is quite weak due to its $1/N^2$ dependence, the low values we obtain are a consequence of considering only off-resonant Raman scattering (see Fig. 6a), ignoring non-Condon effects, and more importantly, considering a single-cavity mode. We expect $\Gamma_{scatt}$ to increase for multimode cavities since dark state (virtual) emission can be resonantly and off-resonantly scattered from the entire lower polariton branch (the sum over all intermediate states $|\xi, \{n_j\}\rangle$ and final states $|\xi', \{n_j'\}\rangle$ in Eq. (16) must also include a sum over all $k_{\parallel}$ and $k_{\parallel}'$ modes, respectively). Moreover, $\Gamma_{scatt}$ can be enhanced by increasing the field confinement (reducing $N$), although it is not clear if this mechanism is more convenient to measure Raman signals than conventional cavity-enhanced Raman spectroscopy[93]. Yet, resonant scattering from a high-frequency polariton mode at finite $k_{\parallel}$ can assist in the energy transfer into a lower frequency polariton at lower values of $k_{\parallel}$ for which radiative pumping is inefficient due to little overlap with the bare emission spectrum (see Fig. 1). In conclusion, there is no reason to believe that VAS cannot be a more efficient energy transfer mechanism than radiative pumping to pump low-frequency polaritons, despite its $1/N^2$ dependence. However, whether this is the case for the parameter regimes in experiments requires further analysis that will be the focus of future works.

## Discussion

In summary, we have used an exact bosonic mapping of the generalized Holstein–Tavis–Cummings Hamiltonian based on its projection to a subspace of permutationally symmetric vibronic states. The resulting bosonic Hamiltonian describes molecular polaritons for arbitrary internal vibrational structure, number of molecules, and number of excitations[54–58]. Here, we show that this formalism is ideal to study molecular polaritons beyond the $N \to \infty$ limit numerically and analytically. In particular, we use it to study vibrational relaxation and radiative pumping mechanisms. We find that the relaxation mechanism in play is determined by the competition between single-molecule light–matter coupling and weak vibronic couplings, and characterize each mechanism based on their underlying photophysical processes.

We show that radiative pumping is the emission from Stokes-shifted molecules into the polaritons, and can be divided into transmitted and reabsorbed components. The latter leads to a polariton-assisted photon recycling mechanism that allows for long-range energy transfer. On the other hand, we show that vibrational relaxation includes radiative pumping as well as higher-order processes in the single-molecule light–matter coupling $g$; up to second-order processes in the weak linear vibronic coupling regime. We find that each order in $g$ is penalized by a $1/N$ factor in the rate, suggesting that the main contribution to the vibrational relaxation rate comes from radiative pumping. Finally, we classify the second-order processes in $g$ as polariton-assisted Raman scattering, which occurs when the frequency difference between the bare emission and the polariton state coincides with the vibrational excitation created in the Raman process.

Our work constitutes a rigorous derivation and comparison between these polariton relaxation rates, offering a path forward to study molecular polaritons beyond the $N \to \infty$ limit. Finally, since the bosonic formalism already allows arbitrary number of excitations, we believe it is ideal to study processes such as exciton-polariton condensation.

## Methods

### Partitioning the molecular polariton Hamiltonian

We start by partitioning the total bosonic Hamiltonian of Eq. (4) in terms of a zeroth-order Hamiltonian ($\hat{H}_0$), a weak vibronic coupling Hamiltonian ($\hat{H}_{vc}$), and a single-molecule light–matter coupling Hamiltonian ($\hat{H}_{sm}$):

$$
\begin{aligned}
\hat{H} &= \hat{H}_0 + \hat{H}_{vc} + \hat{H}_{sm} \\
\hat{H}_0 &= \omega_c \hat{a}^\dagger \hat{a} + \sum_{i=1}^m \omega_{g,i} \hat{b}_i^\dagger \hat{b}_i + \sum_{i=1}^m \omega_{e,i} \hat{B}_i^\dagger \hat{B}_i \\
&\quad + \sum_{i,j=1}^{m'} V_{eg,ij} \hat{B}_i^\dagger \hat{B}_j + g\left(\hat{B}_1^\dagger \hat{b}_1 \hat{a} + \hat{B}_1 \hat{b}_1^\dagger \hat{a}^\dagger\right) \\
\hat{H}_{vc} &= \sum_{i=1}^{m'} \sum_{j>m'}^m \left(V_{eg,ij} \hat{B}_i^\dagger \hat{B}_j + V_{eg,ji}^* \hat{B}_i \hat{B}_j^\dagger\right) + \sum_{i,j>m'}^m V_{eg,ij} \hat{B}_i^\dagger \hat{B}_j \\
\hat{H}_{sm} &= g \sum_{i>1}^m \left(\hat{B}_i^\dagger \hat{b}_i \hat{a} + \hat{B}_i \hat{b}_i^\dagger \hat{a}^\dagger\right).
\end{aligned}
\tag{17}
$$

This partitioning is based on an important observation: at zero-temperature, all ground-state molecules are in the $\hat{b}_1$ mode, and light–matter coupling is collectively enhanced at the FC configuration via bosonic stimulation. Therefore, the term $g(\hat{B}_1^\dagger \hat{b}_1 \hat{a} + \hat{B}_1 \hat{b}_1^\dagger \hat{a}^\dagger)$ must be included in $\hat{H}_0$, while single-molecule light–matter coupling terms ($\langle \hat{H}_{sm}\rangle \sim g$) can be considered perturbatively (see Supplementary Section 2 for details). On the other hand, the separation of the vibronic coupling terms into strong and weak, given by $m'$, is to some degree arbitrary, and resembles the separation of a total Hamiltonian into a system and a bath in open quantum systems. In general, vibronic coupling terms that can lead to vibronic features in the linear response must be included in $\hat{H}_0$, while weak vibronic couplings away from the FC region, that are small enough to afford a perturbative treatment, should be included in $\hat{H}_{vc}$. Moreover, the definition of $\hat{H}_{vc}$ should be made such that it allows for the definition of physical meaningful relaxation rates. For example, in the problem of triplet harvesting in the collective strong coupling regime[85], defining the weak vibronic coupling Hamiltonian $\hat{H}_{vc}$ as the spin–orbit coupling (singlets are in the interval $i = 1 - m'$ while triplets are in the interval $i = m' - m$) allows for the definition of RISC rates from dark to polaritons states. In quantum optics models where the absorption spectrum showcases only two clear polariton peaks[75], considering all vibronic couplings as weak ($m' = 1$) leads to the definition of vibrational relaxation rates between polaritons and dark states[39,75,87,94]. More formally, the partitioning can be carried out using effective-mode theory[95–97], chain mappings[98,99], variational polaron transformation[85], among other techniques[100].

We assume that the cavity leakage is much faster than each of the aforementioned relaxation rates. Next, we will use the partitioning of the Hamiltonian to show that the competition between $\hat{H}_{vc}$ and $\hat{H}_{sm}$ gives rise to two regimes described by vibrational relaxation ($\langle\hat{H}_{vc}\rangle \ll g$) and radiative pumping ($\langle\hat{H}_{vc}\rangle \gg g$). We interpret each of these relaxation mechanisms into well-known photophysical processes using perturbation theory. Before doing so, we restrict ourselves to the first excitation manifold (the system only has one electronically excited molecule or one photon, but arbitrary number of molecules with phonons). This is an approximation for the linear regime on the interaction between the cavity and the external laser[101]. We also simplify the notation of the many-body states so that they showcase only essential information. This is done by using the multi-particle states introduced by Philpott[73], which has been applied to the polariton system by Herrera and Spano[41], and in our CUT-E formalism[71]:

$$
\begin{aligned}
|1\rangle &= |N00\cdots0, 00\cdots0, 1\rangle \\
|e_k\rangle &= |(N-1)00\cdots0, \cdots1_k\cdots, 0\rangle \\
|g_k1\rangle &= |(N-1)\cdots1_k\cdots, 00\cdots0, 1\rangle \\
|g_k e_{k'}\rangle &= |(N-2)\cdots1_k\cdots, \cdots1_{k'}\cdots, 0\rangle \\
|g_k g_{k'}1\rangle &= |(N-2)\cdots1_k\cdots1_{k'}\cdots, 00\cdots0, 1\rangle \\
|g_k g_{k'} e_{k''}\rangle &= |(N-2)\cdots1_k\cdots1_{k'}\cdots, \cdots1_{k''}\cdots0, 0\rangle.
\end{aligned}
\tag{18}
$$

This approach shares deep connections with previous works by Herrera and Spano[102,103]. Their work considers only one-particle ($|1\rangle$ and $|e_k\rangle$) and two-particle ($|g_k1\rangle$ and $|g_k e_{k'}\rangle$) states, that are either permutationally or non-permutationally symmetric. Our formalism extends this picture by going beyond two-particle states. Although the one- and two-particle states are enough to account for the mechanism of radiative pumping in different regimes of vibronic coupling and collective light–matter coupling strengths[102], accounting for three-particle states and beyond is required in the characterization of polariton photoluminescence. For example, here we show that inclusion of three-particle states ($|g_k g_{k'}1\rangle$ and $|g_k g_{k'} e_{k''}\rangle$) allows for the characterization of a new mechanism we call polariton-assisted Raman scattering, which we believe corresponds to the VAS mechanism experimentally observed by Coles et al.[70]. Although we leave out non-permutationally symmetric states in this work, we argue that they cannot be populated if the initial state is permutationally symmetric.

## Numerical simulations

We consider a chromophore described by a two-mode linear vibronic coupling Hamiltonian[104],

$$\hat{H}_m = \sum_{i=1}^{2} \omega_{\nu,i}\hat{\beta}_i^\dagger \hat{\beta}_i + \left[ \omega_0 + \sum_{i=1}^{2} \omega_{\nu,i}\sqrt{s_i}\left(\hat{\beta}_i^\dagger + \hat{\beta}_i\right) \right] |e\rangle\langle e|, \quad (19)$$

where $\hat{\beta}_i$ annihilates a phonon in the vibrational mode $i$th, with frequency $\omega_{\nu,i}$, and vibronic coupling determined by the Huang-Rhys (HR) factor $s_i$. We choose the molecule to have strong vibronic coupling to a high and a low-frequency modes (see Supplementary Fig. 2 for PESs).

For radiative pumping, the parameters chosen were $\omega_{\nu,1} = 12.5\omega_{\nu,2} = 0.01$ a.u. In Fig. 4, we calculate the radiative pumping rate for $\sqrt{s_1} = 1$, $\omega_c = \omega_0 + \omega_{\nu,1}s_1 + \omega_{\nu,2}s_2$ (cavity resonant with the molecular vertical transition), $g\sqrt{N} = 0.04$ a.u., $N = 10^5$ molecules, $\gamma_\xi = \gamma = 0.0015$ a.u., and different values of $\sqrt{s_2}$.

For polariton-assisted Raman scattering, we chose parameters $\omega_{\nu,1} = 10\omega_{\nu,2} = 0.01$ a.u., $\sqrt{s_1} = 0.3$, $\omega_c = \omega_0 + \omega_{\nu,1}s_1 + \omega_{\nu,2}s_2$, $g\sqrt{N} = 0.035$ a.u., $N = 10^5$ molecules, $\gamma_\xi = \gamma = 0.0015$ a.u., and different values of $\sqrt{s_2}$.

Notice that the computational cost of our calculations does not scale with the number of molecules.

## Data availability

Data sharing is not applicable to this article as no datasets were generated or analyzed during the current study. The plots in Figs. 3 and 6 result from the numerical evaluation of Eqs. (9) and (16), respectively.

## Code availability

The computational code used to generate the plots in the present article is available by email upon request to the authors.

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

## Acknowledgements

J.B.P.S. and J.Y.Z. were funded by the Air Force Office of Scientific Research (AFOSR) through the Multi-University Research Initiative (MURI) program no. FA9550-22-1-0317. The authors also thank Arghadip Koner, Kai Schwennicke, Stéphane Kéna-Cohen, Gerrit Groenhof, and Jonathan Keeling for useful discussions.

## Author contributions

J.B.P.S. and J.Y.Z. conceived the project and participated in the discussions. J.B.P.S. conducted theoretical and numerical studies. J.B.P.S. wrote the manuscript with input from J.Y.Z.

## Competing interests

The authors declare no competing interests.
