## [Transparent Peer Review file · Nature Communications]

Radiative pumping vs vibrational relaxation of molecular polaritons: a bosonic mapping approach

Corresponding Author: Professor Joel Yuen-Zhou

Version 0:

Reviewer comments:

Reviewer #1

(Remarks to the Author)

Pérez-Sánchez et. al. develop a theoretical model for studying molecular exciton-polariton relaxation rates induced by anharmonic vibrations. They first start with an anharmonic version of the Holstein-Tavis Cummings model that includes, 1 cavity mode, N identical molecules with one vibrational mode on each molecule (N vibrations). Then, they write down the TDSE for the entire system, and using the fact that all molecules are identical (i.e. using permutation symmetry of the coefficients of wavefunction), they derive an effective Hamiltonian that they then use to study the relaxation process. It turns out that each successive excitation space is like a state of a harmonic oscillator, and thus they use Bosonic creation annihilation operators to rewrite a compact light-matter Hamiltonian. Then they use simple Fermi's Golden Rule to investigate two scenarios: (a) when vibronic couplings are high compared to single-molecule light-matter couplings and when single-molecular light-matter coupling is higher. They also study a chromophore system and show that photons absorbed by one molecule are reabsorbed by other molecules, which they term photon recycling.

I think the mathematical aspect of this work is intellectually interesting. However, this formalism of exploiting permutation symmetry is already published by the authors (C-UJE), thus this work essentially looks like an extension to their previous work. The results are qualitatively expected, and I have many concerns with the results presented. Thus, unfortunately, I cannot recommend the paper for publication at Nat. Comm. Below are my specific comments:

* Minor: It is not clear to me what was the value or the precise definition W used in this work.

* Overall, to me it looks like that this approach is similar to the development of MCTDHB (<https://journals.aps.org/pr/abstract/10.1103/PhysRevLett.99.030402>) - I would appreciate, if the authors could clarify if this is so.

* Importantly, I think Fig.1 is deceptive. This work specifically works with a single cavity mode, thus drawing the polariton dispersion is not correct. The arrows indicating relaxation over the polariton band simply do not exist in the model studied here.

* This brings me to one of the major drawbacks of the present work. Single cavity mode approximation. Polariton relaxation processes involve Fröhlich scattering-like processes (e.g. <https://doi.org/10.1063/5.0037868>) not included in this work. Which probably will be as important or more important than the scattering process studied here. Thus, I don't know to what extent we can rely on the results here with this approximation.

* The author uses the permutation symmetry to generate the Hamiltonian, but then uses it to obtain rates using FGR. Which, in my opinion, defeats the purpose of exploiting the permutation symmetry. For this model, why not perform direct quantum dynamics simulation ? or do redfield-like theory (or lindblad), as was done in <https://doi.org/10.1364/OPTICA.5.001247>. Also, doesn't this work (<https://doi.org/10.1364/OPTICA.5.001247>) simulate the same model as in here (esp. comparing the

chromophore system + bath) ?

* If the formulation is exact, can the authors clarify if they can do direct dynamics of the model Hamiltonian they derived ? Would simulating it directly capture the effects of dissipation by the environment/vibration (thus not requiring the need of using any master-equation like approach)? Its not very clear to me if this is so, since the final Hamiltonian involves few Bosonic modes and as a result I expect to get recurrences in the population dynamics (if that is so - does that mean we cannot capture dissipative effects once the permutation symmetry has been applied).

* I think the term "generalized Tavis-Cummings" is not appropriate to use. In the literature (e.g. <https://www.annualreviews.org/content/journals/10.1146/annurev-physchem-010920-102509>) it is used for referring to the multimode dipole gauge Hamiltonian.

Reviewer #2

(Remarks to the Author)

Pérez-Sánchez and Yuen-Zhou propose a theoretical model based on the bosonic mapping on the molecular vibronic states in exciton polaritons. The authors claim that the formalism proves effective for analyzing molecular polaritons in the finite N regime, particularly for vibrational relaxation and radiative pumping mechanisms, which depend on the balance between single-molecule light-matter coupling and weak vibronic couplings, with radiative pumping significantly influencing the relaxation rate.

The theoretical framework presented in this manuscript offers an intriguing perspective for configuring the entire molecular exciton strong coupling structure and may significantly influence experimental design for relevant systems. Therefore, I would recommend the publication of this manuscript in Nature Communications after the following questions are adequately addressed:

1. On page 3, the authors treat the weak vibronic coupling and the single-molecule light-matter coupling (g) as two competitive regimes. However, the vibrational excitation on electronic ground state would change the transition dipole moment in the FC regime, which would directly modify the coupling strength. Why do the authors claim the vibronic coupling is weak enough to be away from that and decouple the H_{vc} (W) and H_{sm} (g)? Do the authors aim for specific molecular vibronic structures in the model?
2. In part D on page 11, do the authors consider more than one photons couple to the single-molecule vibronic state? If so, would the coupling strength modulate as a factor of $\sqrt{n\phi+1}$?
3. In the polariton-assisted photon emission and re-absorption processes, why the resonant energy transfer mechanism can be ignored? Is the transition from $|ss\rangle$ to $|ss'\rangle$ possible in Fig. 4b without the assistant of polariton state $|\xi\rangle$?
4. The photon recycling would be possible to create a continuous feeding of bosonic population at finite $k_{||}$. How much polariton-polariton interaction at smaller $k_{||}$ would influence the radiative pumping to the LP state?

Reviewer #3

(Remarks to the Author)

In the context of exciton polaritons that emerge from the strong interaction between molecular excitations and confined photons, the authors derive from a microscopic Holstein-Tavis-Cummings Hamiltonian the two main relaxation mechanisms that have been proposed for these hybrid light-matter quasi-particles: Radiative Pumping (RP) and Vibrationally Assisted Scattering (VAS). Relying on a well established approach to map this Hamiltonian for identical permutationally symmetric few-level systems into boson operators, they derive expressions for the rates of both mechanisms (RP and VAS) employing Fermi's Golden Rule. Results for numerical simulations illustrate how this approach can be employed to investigate molecular polaritons in the large N limit, typical of experimental realisations of exciton polariton condensation and lasing that build upon polariton relaxation.

This work builds up on previous theoretical developments (references 32,33,60,61,63,64,81) aimed at investigating the relaxation of polaritons in disorder materials with the key difference that it separates RP and VAS by respectively treating photonic and vibronic couplings of the Hamiltonian as first order perturbations. With this strategy, Juan B. Pérez-Sánchez and Joel Yuen-Zhou provide a rationale for recent atomistic molecular dynamics simulations that suggested that the same molecular vibrations drive RP and VAS (reference 77). The main finding of the work is however that, for materials with strong electron-vibron coupling, VAS includes additional terms that arise from higher order terms on the light-matter coupling g .

The results are important for both theoreticians and experimentalists in the polariton community and I believe that a derivation from first principles of the rates of both RP and VAS belongs to a journal such as Nature Communications. However, I have a few suggestions for the authors prior publication.

Comment 1:

The manuscript is well written and the literature related to this work is properly cited. Nevertheless, a few citations are still missing, in particular to guide a general readership less familiar with polaritons. Therefore, I suggest the authors to add work of their choice referring to single molecule strong coupling in page 2, line 26, collective strong coupling in page 2, line 27,

the large N problem in page 3, line 50 (besides the work in preparation they make reference to) and the bosonization approach in page 3, line 58 (the authors can here already cite references 47 to 51 so that it is clear that this is not the novelty of the presented work).

Comment 2:

Separating RP and VAS by respectively treating photonic and vibronic couplings of the Holstein- Tavis-Cummings Hamiltonian as first order perturbations is a well justified approach since the community has considered these mechanisms as radiative and non-radiative relaxation channels, respectively. While the authors admit that the choice of m' for partitioning the Hamiltonian is arbitrary, it would be highly desirable that they provide further guidelines on how to choose it with respect to the system parameters, in particular the light-matter and vibronic couplings.

Comment 3:

In connection to the previous comment, the authors claim that their work generalizes the formalism proposed by Herrera and Spano (references 60 and 61). In order to better show the connection to that work, could the authors identify the same three light-matter coupling regimes described in reference 60 for a material of their choice and comment on whether RP or VAS will constitute the main relaxation mechanism?

Comment 4:

In page 7, line 150, the authors provide an equation to define the first excitation manifold. While for a familiar readership such manifold is obvious, in my opinion the provided equation brings confusion because of the sum over the m vibrational modes. I would suggest a definition making explicit the fact that at most there is one excitation in the system: either there is a photon and all molecules are in their ground state, either one and only one molecule is in its excited state and all other are in the ground state. Related to this: I understand the maximum number of allowed vibrations is $N \times m$. Is that correct?

Comment 5:

Concerning the polariton-assisted Raman scattering (section V, pages 15-16), does the model presented by the authors suggest that polaritons could lead to amplified Raman scattering signals?

Technical suggestions:

The paper could be further improved by relating more strongly the figures to the text. Along the same line, in my opinion, the right panel of Figure 1 does not illustrate in a clear manner the relaxation mechanisms reported in this work.

Reference 37 is pointing to the arXiv version of that work while that work has already been accepted in a peer reviewed journal i.e. Dutta et al., Nat Commun 15 (2024) 6600

The name of the first author of reference 59 is incorrect. It is not Martínez but Sánchez Martínez.

Page 13, line 264: there is a typo in the name of the first author of references 31, 64 and 65.

Version 1:

Reviewer comments:

Reviewer #1

(Remarks to the Author)

The authors have clarified most of my questions. However, I remain somewhat skeptical about the publication of this draft in its present form in Nature Communications for the following reasons:

1. It is not at all clear, to what extent the single-mode approximation remains valid, especially for calculating such a relaxation rate given the absence of Frohlich-like scattering.

The authors state, "our results can be extended to the multimode scenario owing to a crucial observation by Engelhardt and Cao [90]: the coupling between different cavity $k||$ modes vanishes in the limit where $N \rightarrow \infty$ ". This statement is wrong/misrepresentation of what has been discussed in Ref. 90.

In the work of Engelhardt and Cao [90], a multimode disordered Tavis-Cummings model (which does not include phonons) has been considered, where the Heisenberg equation of motion of the cavity modes [see SI of Ref. 90] with different k 's are coupled due to their coupling to the matter degrees of freedom whose onsite energies have Gaussian disorder. It is this coupling that vanishes in the thermodynamic limit. This has nothing to do with the phonon-induced scattering between polariton states (of the same branch), as phonon-induced (dynamic) disorder, even in the semi-classical picture, cannot be simply modeled with static disorder. Therefore, it is incorrect to state that phonon-induced Frohlich-like scattering between polariton states at different k 's is unimportant because a work finds it so when ignoring phonons.

2. I therefore fully disagree with the author's decision to keep Fig.1, in its present form that illustrates a polaritonic dispersion that is fully absent from the theoretical model considered here. In fact, it is because the polariton community is familiar with the polariton dispersion, this figure will cause confusion, as seeing this would lead one to expect the work and the analytical

expression derived to have considered the Frohlich-like scattering processes. I am sorry, but I cannot recommend this work unless this figure is made accurate.

Reviewer #2

(Remarks to the Author)

Pérez-Sánchez and Yuen-Zhou have thoroughly addressed my previous questions and comments, particularly regarding the various contributions in a complex polariton system involving multimode interactions. The revised manuscript further clarifies the independence between vibronic coupling and light-matter interactions in the finite-N molecule regime, as well as the role of cavity photons in the strongly coupled photon-cycling scheme. Additionally, the authors have expanded the theoretical description to encompass a broader range of k-values, potentially linking it to the Bose scattering mechanism. Overall, the authors' responses are satisfactory, and I recommend publication in Nature Communications.

Reviewer #3

(Remarks to the Author)

The authors have satisfactorily addressed most of my comments and the manuscript has significantly gained in both clarity and position within the field despite that the Figures remain poorly related to the text. I have two additional minor comments prior publication:

Comment 1:

With the increased clarity of the work and the lengthier description of the so-called "photon-recycling" mechanism, I now understand that this corresponds to population exchanges between dark and bright states mediated by photons that get emitted and re-absorbed in the polariton system. Since the authors additionally stress that this mechanism is responsible for long range polariton transport, I believe work dedicated exclusively to the study of polariton propagation and showing the importance of this same mechanism, although differently labelled, should be cited i.e. Sokolovskii et al., Nat. Commun. 14 (2023) 6613 and Balasubrahmaniam et al. Nat. Materials 22 (2023) 338.

Comment 2:

Although in the original version of the manuscript, the authors had already drawn attention on some of the possible limitations of their model because of the single cavity mode approximation, comment 5 of Reviewer 1 is of outmost importance.

If at present it is not possible to account for the entire cavity description, the authors should at least tone down their message by avoiding strong statements as the one in page 8, lines 167-169 "accounting for three-particle states and beyond is important for a complete characterization of polariton photoluminescence" when polariton photoluminescence, specially in the low excitation limit, cannot actually be "completely characterized" because of the single mode approximation.

Also, concerning comment 4 of Reviewer 1, while I understand that Figure 1 is an illustrative figure with which the polariton community is familiar owing in particular to reference 67 [Coles et al. Adv. Funct. Mater. 21 (2011) 3691], the authors should stress in the caption of that figure that the entire polariton dispersion is not accounted for in the present work.

Version 2:

Reviewer comments:

Reviewer #1

(Remarks to the Author)

I have reviewed the modified draft. The authors have addressed all of my concerns. I recommend the publication of this work.

**Radiative pumping vs vibrational relaxation of molecular
polaritons: a bosonic mapping approach**

Answer to reviewers

Juan B. Pérez-Sánchez and Joel Yuen-Zhou*

Department of Chemistry, University of California San Diego, La Jolla, CA 92093, USA

(Dated: November 14, 2024)

In black: questions and comments by reviewers.

In blue: answer to reviewers' comments.

In red: modifications of the manuscript addressing reviewer's comments. They also appear highlighted in red in the manuscript file.

I. REVIEWER 1

Pérez-Sánchez et. al. develop a theoretical model for studying molecular exciton-polariton relaxation rates induced by anharmonic vibrations. They first start with an anharmonic version of the Holstein-Tavis Cummings model that includes, 1 cavity mode, N identical molecules with one vibrational mode on each molecule (N vibrations). Then, they write down the TDSE for the entire system, and using the fact that all molecules are identical (i.e. using permutation symmetry of the coefficients of wavefunction), they derive an effective Hamiltonian that they then use to study the relaxation process. It turns out that each successive excitation space is like a state of a harmonic oscillator, and thus they use Bosonic creation annihilation operators to rewrite a compact light-matter Hamiltonian. Then they use simple Fermi's Golden Rule to investigate two scenarios: (a) when vibronic couplings are high compared to single-molecule light-matter couplings and when single-molecular light-matter coupling is higher. They also study a chromophore system and show that photons absorbed by one molecule are reabsorbed by other molecules, which they term photon recycling.

We would like to respectfully clarify that the reviewer does not correctly summarize our work. First, although our work can model anharmonic vibrations and it is particularly useful when such vibrations are strongly coupled to the electronic transitions, we do not only study relaxation mechanisms induced by anharmonic vibrations. Instead, we study relaxation mechanisms called radiative pumping and vibrational relaxation, the first one is a mechanism driven by single-molecule light-matter coupling, while the latter is driven by vibronic coupling. Second, our formalism is not restricted to a single vibrational mode per molecule. Instead, we include an arbitrary number of vibrational modes per molecule, making our formalism completely general regarding the molecular model. We clarify this in the manuscript when we write " q_i is the vector of all vibrational degrees of freedom of

* joelyuen@ucsd.edu

molecule i ".

Finally, we believe the reviewer might be missing some of the main results of our work. 1) We derived for the first time a completely general rate for radiative pumping and characterize it simply as fluorescence from incoherent excitons (vibronic dark states). This rate is written in terms of polariton linear optical properties so that experimentalists can easily use it without having to perform any quantum chemistry calculations (the formula is independent of the molecular model). 2) We demonstrated that vibrational relaxation is equivalent to radiative pumping in the weak linear vibronic coupling limit. This is important because there has been confusion for decades (since the work by Litinskaya in 2004) as to what exactly is vibrational relaxation. 3) We characterized and provide a rate for polariton-assisted photon recycling, which is a strong coupling effect responsible, among other things, for long range energy transfer in polaritonic systems. This rate is also given in terms of polariton linear optical properties. 4) We discovered a novel mechanism called polariton-assisted Raman scattering, which provides an explanation for the mechanism called vibrationally-assisted scattering observed back in 2011 by Coles et al., which remains unexplained. 5) Finally, our work provides definitions of basic concepts of molecular polaritons such as vibronic polaritons and dark states, makes comparisons with definitions based on quantum optics models, and makes connections with other important works in the field.

Although the reviewer accurately describes the mathematical methods used in this work, these do not constitute the main contribution of this work. In fact, we let that clear when we write:

“In our previous work we have derived Collective Dynamics Using Truncated-Equations (CUT-E), a formalism that, by exploiting the permutational symmetries of the exact time-dependent many-body (many-molecule and cavity) wavefunction, yields a hierarchy of timescales that renders the simulation efficient for large N [45]. Here, we recognize that this formalism can be easily derived using second quantization.”

“The bosonic mapping of identical-noninteracting particles is well known [38, 39, 46], and it has been applied to ensembles of d -level systems strongly interacting with light [47–51] (also see Reference 52 for a fermionic mapping). We carry out this mapping by...”

We have modified a paragraph in the introduction to clarify further that the bosonic mapping itself is not the main novelty of this work:

In page 13 we write “The aim of this work is twofold: first, we re-derive an exact bosonic

picture of organic molecular polaritons from first principles to study molecular dynamics under collective light-matter coupling for arbitrary number of molecules N , excitations N_{exc} , and internal vibrational structure of the molecules [51–57]. This mapping is analogous to the Schwinger boson representation of spins [58], and is ideal for numerical simulations using Meyer-Miller mappings [59–62], quantum cumulant expansions [63], quantum computing with bosonic devices [64], and other quantum mechanical methods suited for bosonic systems [65]”

Yet, we would like to clarify that the mapping of the Hamiltonian to the bosonic picture and the use of the bosonic picture to define relaxation rates are two completely different problems. While the former is not a novelty of our work, the latter certainly is.

In summary, while the bosonic mapping is not completely new, the conceptual finds are. Since this confusion can be due to us not successfully explaining this in the introduction of the manuscript, we have modified it to make our main contributions clearer.

In page 3 we now write: “In this work we re-derive an exact bosonic picture of organic molecular polaritons from first principles to study molecular dynamics under collective light-matter coupling for arbitrary number of molecules N , excitations N_{exc} , and internal vibrational structure of the molecules [51–57]. This mapping is analogous to the Schwinger boson representation of spins [58], and is ideal for numerical simulations using Meyer-Miller mappings [59–62], quantum cumulant expansions [63], quantum computing with bosonic devices [64], and other quantum mechanical methods suited for bosonic systems [65]. Next, we focus on the large (yet finite) N case and the first excitation manifold to rigorously derive radiative pumping and vibrational relaxation rates. We achieve this by partitioning the bosonic Hamiltonian into “fast” and “slow” components, treating the slow components perturbatively. Next, we unambiguously establish the fundamental differences between these two polariton relaxation mechanisms. Radiative pumping is the emission from incoherent excitons that populate the polaritons, which can either leak out of the cavity or be reabsorbed by the material. The latter, which we call polariton-assisted photon recycling, is a strong coupling phenomenon responsible for long-range energy transfer and may play an important role in polariton transport and polariton condensation. We also provide simple analytical formulas for radiative pumping and polariton-assisted photon recycling in terms of linear optical properties, which can be easily used by experimentalists without the need of quantum chemistry calculations. On the other hand, vibrational relaxation includes radiative pump-

ing and higher-order processes in the single-molecule coupling g such as polariton-assisted Raman scattering, which we believe corresponds to vibrationally-assisted scattering (VAS). We lay down approximations to calculate polariton-assisted Raman scattering rates, which we apply on a simple model. In Fig. 1 we summarize the molecular polariton photophysics discussed in this manuscript”

In the conclusions we write: “The latter leads to a polariton-assisted photon recycling mechanism that allows for long-range energy transfer”.

1) I think the mathematical aspect of this work is intellectually interesting. However, this formalism of exploiting permutation symmetry is already published by the authors (C-UTE), thus this work essentially looks like an extension to their previous work. The results are qualitatively expected, and I have many concerns with the results presented. Thus, unfortunately, I cannot recommend the paper for publication at Nat. Comm. Below are my specific comments:

We disagree with the reviewer that our work should be assessed based on the mathematical methods used, as these are not the main novelty of our work (we explain this at length in the first comment). Instead, it should be assessed based on the relevance of the findings (we explain them in detail in the previous comment as well). In summary, the importance of our work lies on the answers it provides to long-standing questions in the field such as the definition of polariton and dark states, the nature of radiative pumping, the difference between radiative pumping and vibrational relaxation, the mechanism behind long-range energy transfer, the formal derivation for relaxation rates for arbitrary molecular systems in terms of linear optics quantities, and the existence of the previously observed phenomenon called vibrationally-assisted scattering, that we call polariton-assisted Raman scattering.

We particularly disagree with the statement “The results are qualitatively expected”, as these results are either not known to the community (regarding photon recycling and Raman scattering) or no formal derivation of the rates from first principles exists (radiative pumping).

We respectfully encourage the reviewer to re-evaluate the significance of our work beyond the mathematical methods involved and the corrections we have made to the manuscript based on their comments and those from other reviewers.

2) Minor: It is not clear to me what was the value or the precise definition W used in this work.

See edited manuscript page 7, line 138: $\langle \hat{H}_{vc} \rangle \sim W$). W is the weak-vibronic coupling that is small enough to afford a perturbative treatment. In general, W must be smaller than all the fast timescales of the system such as vibronic couplings that give rise to vibronic progressions or the collective coupling strength. This is to allow for the use of techniques like first-order perturbation. Although there are many ways to partition the Hamiltonian to meet this requirement, ideally W must have a physical meaning. For example, in the problem of triplet harvesting in the collective strong coupling regime, the magnitude of W would be that of the spin orbit coupling that allows for reverse intersystem crossing.

We realized we may have not explained this very clearly in the original manuscript. We have modified the paragraph below Eq. 5 to explain this better. We write: “This partitioning is based on an important observation: at zero-temperature, all ground-state molecules are in the \hat{b}_1 mode, and light-matter coupling is collectively enhanced at the FC configuration via bosonic stimulation. Therefore, the term $g(\hat{B}_1^\dagger \hat{b}_1 \hat{a} + \hat{B}_1 \hat{b}_1^\dagger \hat{a}^\dagger)$ must be included in \hat{H}_0 , while single-molecule light-matter coupling terms ($\langle \hat{H}_{sm} \rangle \sim g$) can be considered perturbatively (see Appendix B for details). On the other hand, the separation of the vibronic coupling terms into strong and weak, given by m' , is to some degree arbitrary, and resembles the separation of a total Hamiltonian into a system and a bath in open quantum systems. In general, vibronic coupling terms that can lead to vibronic features in the linear response must be included in \hat{H}_0 , while weak vibronic couplings away from the FC region, that are small enough to afford a perturbative treatment, should be included in \hat{H}_{vc} ($\langle \hat{H}_{vc} \rangle \sim W$). Moreover, the definition of \hat{H}_{vc} should be made such that it allows for the definition of physical meaningful relaxation rates. For example, in the problem of triplet harvesting in the collective strong coupling regime [70], defining the weak vibronic coupling Hamiltonian \hat{H}_{vc} as the spin-orbit coupling (singlets are in the interval $i < m'$ while triplets are in the interval $i = m' - m$) allows for the definition of RISC rates from dark to polaritons states. In quantum optics models where the absorption spectrum showcases only two clear polariton peaks [71], we will show that considering all vibronic couplings as weak ($m' = 1$) leads to the definition of vibrational relaxation rates between polaritons and dark states [37, 71–73]. More formally, the partitioning can be carried out using effective-mode theory [74–76], chain mappings [77, 78], variational polaron transformation [70], among other techniques [79].”

3) Overall, to me it looks like that this approach is similar to the development of MCTDHB (<https://journals.aps.org/prl/abstract/10.1103/PhysRevLett.99.030402>) - I would appreci-

ate, if the authors could clarify if this is so.

We expect the mathematics used in the development of MCTDHB (MCTDH for bosons) to be similar to the one we used in this work because non-interacting molecules behave like bosonic particles when restricted to the permutationally-symmetric subspace. In fact, in page 3 we pointed out that MCTDHB can be used to perform efficient quantum dynamics calculations of molecular polaritons, in particular when arbitrary number of excitations are present. However, it is important to clarify that, in the calculation of the relaxation rates, we do not use time-dependent single-particle functions (as is characteristic in MCTDH).

We thank the reviewer for pointing us to that reference. We have included it in the manuscript. In page 3 we now write: "... This mapping ... is ideal for numerical simulations using ... the multi-configurational time-dependent Hartree for bosons (MCTDHB) [65,66]."

4) Importantly, I think Fig. 1 is deceptive. This work specifically works with a single cavity mode, thus drawing the polariton dispersion is not correct. The arrows indicating relaxation over the polariton band simply do not exist in the model studied here.

We agree with the reviewer that our work considers a single cavity mode. This means that the multimode representation of the relaxation mechanisms we characterize is, in principle, not justified. However, we originally decided to include it for two reasons: 1) it is more familiar to the polaritonics community (e.g., reviewer 3 prefers it over Fig. 1 right panel). 2) The only additional assumption in that figure is that relaxation from dark states via single-molecule light-matter coupling can occur into any polariton $k_{||}$ mode (single-molecule coupling couples different $k_{||}$ modes).

We mentioned this in page 9 of the original manuscript: "Although we cannot explicitly show it here due to the single-mode description of the cavity, radiative pumping can pump any polariton mode that is resonant with the molecular emission in multimode cavities". However, we did not provide any argument for it. We now provide a more detailed justification for this, which we elaborate in the answer to the main comment.

We decided to keep Fig. 1 left panel if our arguments on how to extend relaxation rates to multimode cavities is satisfactory (see next comment).

5) This brings me to one of the major drawbacks of the present work. Single cavity mode approximation. Polariton relaxation processes involve Fröhlich scattering-like processes (e.g. <https://doi.org/10.1063/5.0037868>) not included in this work. Which probably will be as

important or more important than the scattering process studied here. Thus, I don't know to what extent we can rely on the results here with this approximation.

Although we agree with the reviewer that a single-mode description of the relaxation mechanisms cannot account for scattering processes that mix different cavity k_{\parallel} modes, we disagree that they are more important than the ones we characterize in this work. We now explain why radiative pumping and polariton-assisted Raman scattering can be easily extended to the multimode picture, resulting in a description of polariton relaxation in multimode cavities that is consistent with the picture we draw in Fig. 1 left panel. No other scattering mechanisms seem to be left out.

Radiative pumping in multimode cavities:

In page 11 we write: “Although we have employed a single-mode description of the optical cavity to derive equation (11), our results can be extended to the multimode scenario owing to a crucial observation by Engelhardt and Cao [90]: the coupling between different cavity k_{\parallel} modes vanishes in the limit where $N \rightarrow \infty$. This implies that k_{\parallel} is a good quantum number for the polaritonic eigenstates of $\hat{H}_{rp}^{(0)}$, which is equal to the total Hamiltonian in the $N \rightarrow \infty$ limit [42]. This allows us to define linear optical properties $D_{k_{\parallel}}^{(R)}(\omega)$, $A_{k_{\parallel}}(\omega)$, and $T_{k_{\parallel}}(\omega)$. For finite N , the single-molecule coupling \hat{H}_{sm} that gives rise to radiative pumping must couple different cavity modes.”

“Therefore, we postulate that the radiative pumping rate in the multimode case is given by

$$\Gamma_{rp} = \frac{2g^2}{\kappa} \sum_{k_{\parallel}} \int d\omega \sigma_{em}(\omega) \left[A_{k_{\parallel}}(\omega) + 2T_{k_{\parallel}}(\omega) \right],$$

where we sum over all the k_{\parallel} modes. This expression is consistent with the fact that radiative pumping is simply fluorescence from incoherent excitons, and can pump any polariton mode”.

Polariton-assisted Raman scattering in multimode cavities:

In page 18 we write: “We expect Γ_{scatt} to increase for multimode cavities since dark state (virtual) emission can be resonantly and off-resonantly scattered from the entire lower polariton branch (the sum over all intermediate states $|\xi, \{n_j\}\rangle$ and final states $|\xi', \{n'_j\}\rangle$ in equation (19) must also include a sum over all k_{\parallel} and k'_{\parallel} modes, respectively)”

6) The author uses the permutation symmetry to generate the Hamiltonian, but then uses it to obtain rates using FGR. Which, in my opinion, defeats the purpose of exploiting the permutation symmetry. For this model, why not perform direct quantum dynamics simulation ?

or do redfield-like theory (or lindblad), as was done in <https://doi.org/10.1364/OPTICA.5.001247>. Also, doesn't this work (<https://doi.org/10.1364/OPTICA.5.001247>) simulate the same model as in here (esp. comparing the chromophore system + bath) ?

Although we agree that the calculation of FGRs can be done without the use of permutational symmetries in the Hamiltonian, we still consider it valuable to do the calculation in this picture as it provides intuition on how some of these relaxation processes can be modified by phenomena such as bosonic stimulation in higher excitation manifolds. For example, the intuition provided by the bosonic picture recently led us to postulate that long-range energy transfer can be enhanced when many excitations are present in the cavity.

In page 13 we added: “Although we derived this rate in the first excitation manifold, we expect it to be enhanced via bosonic stimulation if many polaritons are present in the mode that mediates the energy transfer process. Despite this effect being analogous to stimulated emission outside of the cavity, there is a unique opportunity in the strong coupling regime, which is to use it to enhance long-range energy transfer to other molecules of the ensemble.”

Although we can perform quantum dynamics simulations of this phenomena, we believe the derivation of relaxation rates can be more useful as they can be directly obtained from spectroscopic measurements.

On the other hand, while we agree that simulating the model with Redfield theory as in <https://doi.org/10.1364/OPTICA.5.001247v> is a valid approach in the weak linear vibronic coupling regime (the Litinskaia rate is derived under similar premises). However, this is not valid in the more general case when vibronic coupling is strong enough to give rise to non-markovian dynamics (which would be reflected in the appearance of vibronic-polariton peaks in the linear response, as opposed to simple two polariton peaks). One of the novelties in our work is that the formulas we derive for the relaxation rates are completely general and do not rely on any kind of approximation to the vibronic coupling.

In page 7 we write: “the separation of the vibronic coupling terms into strong and weak, given by m' , is to some degree arbitrary, and resembles the separation of a total Hamiltonian into a system and a bath in open quantum systems. In general, vibronic coupling terms that can lead to vibronic features in the linear response must be included in \hat{H}_0 , while weak vibronic couplings away from the FC region, that are small enough to afford a perturbative treatment, should be included in \hat{H}_{vc} ($\langle \hat{H}_{vc} \rangle \sim W$)” “In quantum optics models where the absorption spectrum showcases only two clear polariton peaks [71], considering all vibronic

couplings as weak ($m' = 1$) leads to the definition of vibrational relaxation rates between polaritons and dark states [37, 71–73]”

We would like to thank the reviewer for making us aware of this reference. We have now included it in the manuscript.

7) If the formulation is exact, can the authors clarify if they can do direct dynamics of the model Hamiltonian they derived? Would simulating it directly capture the effects of dissipation by the environment/vibration (thus not requiring the need of using any master-equation like approach)? Its not very clear to me if this is so, since the final Hamiltonian involves few Bosonic modes and as a result I expect to get recurrences in the population dynamics (if that is so - does that mean we cannot capture dissipative effects once the permutation symmetry has been applied).

We completely agree with the reviewer that capturing dissipative effects while performing quantum dynamics simulations (for example, starting with a photon in the cavity and the molecules in the global ground state), using the bosonic polariton Hamiltonian, is challenging because we must incorporate many bosonic modes (vibronic states). However, this is not a challenge in this work because the complex excited-state molecular dynamics is encoded in the linear response formulas for polariton absorption, reflection, and transmission. This bypasses the need to calculate the quantum dynamics to obtain the polariton relaxation rates.

To clarify this in the manuscript, in page 10 we added: “Notice that evaluating equation (11) does not require expensive quantum dynamics simulations or quantum chemistry calculations, as all the information about the excited state molecular dynamics is encoded in the polariton linear response formulas, which are routinely measured”

If one is still interested in performing quantum dynamics simulations using the bosonic polariton Hamiltonian in its current form, one must overcome the challenge of having to add many bosonic modes to capture dissipation. This challenge does not arise because of applying permutational symmetries, and we believe a separation of the Hamiltonian into a system bath should be possible, followed by the derivation of master equations. In fact, these permutational symmetries have been used before on top of master equation approaches before (<https://journals.aps.org/pr/abstract/10.1103/PhysRevA.105.043704>, cited in the manuscript). Another approach to deal with the issue of dissipation are semiclassical methods that replace bosonic modes associated to low frequency vibrational excitations with

classical variables via $\hat{B}_i, \hat{B}_i^\dagger \rightarrow Q_i, P_i$, or deriving a kinetic rate equation models from the bosonic Hamiltonian (<https://journals.aps.org/prl/pdf/10.1103/PhysRevLett.111.100404>). We are currently exploring following one of these approaches to study polariton condensation, so we prefer not to discuss it in the present manuscript. Regardless, in future work, some algorithmic development will be required to make the bosonic Hamiltonian computationally useful to carry out quantum dynamics simulations, but this is an effort that is beyond the scope of the current work.

8) I think the term "generalized Tavis-Cummings" is not appropriate to use. In the literature (e.g. <https://www.annualreviews.org/content/journals/10.1146/annurev-physchem-010920-102509>) it is used for referring to the multimode dipole gauge Hamiltonian.

We thank the reviewer for making us aware of this issue of naming the Hamiltonian a "generalized Tavis-Cummings". We agree that this will create confusion in the community. In page 3 we now write "The Tavis-Cummings Hamiltonian, extended to include internal vibrational degrees of freedom missing from original models, can be written as"

We thank the reviewer for their useful comments. We believe we have addressed all their concerns.

II. REVIEWER 2

Pérez-Sánchez and Yuen-Zhou propose a theoretical model based on the bosonic mapping on the molecular vibronic states in exciton polaritons. The authors claim that the formalism proves effective for analyzing molecular polaritons in the finite N regime, particularly for vibrational relaxation and radiative pumping mechanisms, which depend on the balance between single-molecule light-matter coupling and weak vibronic couplings, with radiative pumping significantly influencing the relaxation rate. The theoretical framework presented in this manuscript offers an intriguing perspective for configuring the entire molecular exciton strong coupling structure and may significantly influence experimental design for relevant systems. Therefore, I would recommend the publication of this manuscript in Nature Communications after the following questions are adequately addressed:

We appreciate the reviewer for acknowledging the impact of our work in the polaritonics community.

1) On page 3, the authors treat the weak vibronic coupling and the single-molecule light-

matter coupling (g) as two competitive regimes. However, the vibrational excitation on electronic ground state would change the transition dipole moment in the FC regime, which would directly modify the coupling strength. Why do the authors claim the vibronic coupling is weak enough to be away from that and decouple the $\hat{H}_{vc}(W)$ and $\hat{H}_{sm}(g)$? Do the authors aim for specific molecular vibronic structures in the model?

The reviewer raises the question whether it is valid to treat vibronic coupling and light-matter coupling as independent (in terms of their magnitudes/expectation values). This seems to arise from the possibility of vibronic coupling to cause a reduction of the collective light-matter coupling. This is, however, not correct. We clarify that vibronic coupling only operates in the excited electronic state (it is only responsible for vibrational dynamics in the excited state). This is not an artifact of specific vibronic structures since our formalism is valid for arbitrary potential energy surfaces \hat{V}_g and \hat{V}_e (vibronic coupling is defined by $\hat{V}_{vc} = \hat{V}_e - \hat{V}_g$). Hence, vibronic coupling cannot cause vibrational excitations in electronic ground state molecules that would change the magnitude of the collective light-matter coupling.

On the other hand, we agree that vibrational excitation in the electronic ground state changes the magnitude of the collective coupling. For example, if all molecules are in the global ground state, the collective coupling is $g\sqrt{N}$. However, if 1 molecule is vibrationally excited, the collective coupling becomes $(g\sqrt{N-1})$. These vibrational excitations cannot be created via vibronic coupling (alone) but via light-matter processes such as fluorescence or Raman scattering. This effect is captured in our formalism, although to get analytical formulas for radiative pumping with approximate $g\sqrt{N} \approx g\sqrt{N-1}$, which is valid for large (yet finite) values of N .

In page 5 we added “ $\hat{V}_{eg} = \hat{V}_e - \hat{V}_g$ is the vibronic coupling operator responsible for the molecular dynamics of electronically excited molecules”. In page 10 we added: “More rigorously, $D^{(R)}(\omega)$, $A(\omega)$, and $T(\omega)$ are the linear optical quantities where one of the molecules is vibrationally-excited (molecules emit into polaritons with a slightly reduced Rabi splitting $g\sqrt{N-1}$, an effect negligible in the large N limit)”.

2) In part D on page 11, do the authors consider more than one photons couple to the single-molecule vibronic state? If so, would the coupling strength modulate as a factor of $\sqrt{n_{ph}+1}$?

We thank the reviewer for this question. For this manuscript we limited our analysis to the first excitation manifold (only a single photon/exciton). However, we note that photon

recycling can be stimulated by the number of photons by a factor $\sqrt{N_{ph} + 1}$. This points to a mechanism that enhances polariton-assisted long-range energy transfer rates via bosonic stimulation. However, it would not be the number of photons but the number of polaritons what matters in this case (because the strong coupling between the photons and the matter would form polaritons before the energy transfer mechanism occurs). This mechanism is analogous to stimulated emission outside of the cavity, but the photons emitted under strong coupling can be reabsorbed by the material, while out of the cavity they simply scape.

Because the possibility to enhance long-range energy transfer processes using bosonic stimulation can be of great interest to the polaritonics community, we decided to include this discussion in the manuscript and provide an analytical rate for the polariton-assisted photon recycling mechanism:

In Page 12 we added: “Using Eqs. 11 and 14, the polariton-assisted photon recycling (or energy transfer) rate is given by

$$\Gamma_{rec} = \frac{4g^2\mathcal{Q}}{\kappa} \int d\omega \sigma_{em}(\omega) T(\omega) \text{Im} [\chi^{(1)}(\omega)].$$

Although we derived this rate in the first excitation manifold, we expect it to be enhanced via bosonic stimulation if many polaritons are present in the mode that mediates the energy transfer process. Although this is analogous to stimulated emission outside of the cavity, in the strong coupling regime this can be used to enhance long-range energy transfer to other molecules of the ensemble.”

3) In the polariton-assisted photon emission and re-absorption processes, why the resonant energy transfer mechanism can be ignored? Is the transition from $|ss\rangle$ to $|ss'\rangle$ possible in Fig. 4b without the assistant of polariton state $|\xi\rangle$?

Although in real systems FRET can always occur between sufficiently close molecules, we ignored such mechanism from the beginning when we assumed that molecules in the ensemble do not interact with each other directly (only via the cavity mode) and ignored the optical modes that allow for FRET (see the Hamiltonian in Eq. 1).

In page 12 we added: “Finally, the competition between polariton-assisted photon recycling and Förster resonance energy transfer (FRET, here neglected), and its dependence with the number of polaritons, is an interesting focus of study for incoming works.”

4) The photon recycling would be possible to create a continuous feeding of bosonic population at finite $k_{||}$. How much polariton-polariton interaction at smaller $k_{||}$ would influence

the radiative pumping to the LP state?

Although our formalism does not include a multimode description of the optical cavity, we believe we have a good understanding of how the characterized photophysical mechanisms would behave in the presence of multiple cavity modes.

The radiative pumping will still be dictated by the overlap between the bare molecular emission and the polariton band. This is because radiative pumping is simply fluorescence, and as such, it can pump polaritons with any values of k_{\parallel} , as long as there is spectral overlap. On the other hand, it is typical for photon recycling mechanisms (for example those occurring in luminescent solar concentrators) to result in the emission of photons at lower frequency than those released in the absence of such mechanism. If that is the case for the polariton-assisted photon recycling mechanism as well, then it can result in the pumping of polaritons at lower values of k_{\parallel} , compared to radiative pumping.

Similarly, the polariton-assisted Raman scattering releases a photon at lower frequencies than the one released during fluorescence. However, the efficiency of this mechanism compared to radiative pumping is difficult because the Raman scattering processes depends on the details of the polariton dispersion. This is, in our opinion, one of the most important open questions in the field of polariton photophysics. In the language of polariton-polariton interactions it can be said as follows. Formally speaking, the polariton-assisted photon recycling and the polariton-assisted Raman scattering mechanisms involve polariton-polariton interactions. They are two photon processes where a first photon (virtual or real) is emitted into a polariton mode k_{\parallel} , and a second one can be subsequently released through a lower frequency (although not always) polariton mode k'_{\parallel} . The larger the polariton-polariton interaction the larger are the rates for these two-photon mechanisms. The polariton-polariton interaction is proportional to the single-molecule light-matter coupling strength g , since it vanishes in the limit when $N \rightarrow \infty$ (as shown by Engelhardt and Cao, PRL 130 213602, 2023). However, there are also resonant conditions that play a role in the calculation of the rates. Finally, by definition, radiative pumping is a one-photon process and hence does not involve any polariton-polariton interaction.

In the manuscript we have explained how we believe these mechanisms are modified in the presence of multiple cavity modes. This is based on some of our preliminary results on multimode cavities (not yet published) and the work carried out by Engelhardt and Cao (PRL 130 213602, 2023). In the manuscript we write:

In page 10 we write: “Although we have employed a single-mode description of the optical cavity to derive equation (11), our results can be extended to the multimode scenario owing to a crucial observation by Engelhardt and Cao [90]: the coupling between different cavity k_{\parallel} modes vanishes in the limit where $N \rightarrow \infty$. This implies that k_{\parallel} is a good quantum number for the polaritonic eigenstates of $\hat{H}_{rp}^{(0)}$, which is equal to the total Hamiltonian in the $N \rightarrow \infty$ limit [42]. This allows us to define linear optical properties $D_{k_{\parallel}}^{(R)}(\omega)$, $A_{k_{\parallel}}(\omega)$, and $T_{k_{\parallel}}(\omega)$. For finite N , the single-molecule coupling \hat{H}_{sm} that gives rise to radiative pumping must couple different cavity modes.”

We also now highlight the importance of the polariton dispersions to compare Polariton-assisted Raman scattering and radiative pumping rates to populate the LP at $k_{\parallel} = 0$.

In page 18-19 we write: “We expect Γ_{scatt} to increase for multimode cavities since dark state (virtual) emission can be resonantly and off-resonantly scattered from the entire lower polariton branch (the sum over all intermediate states $|\xi, \{n_j\}\rangle$ and final states $|\xi', \{n'_j\}\rangle$ in equation (19) must also include a sum over all k_{\parallel} and k'_{\parallel} modes, respectively). Moreover, Γ_{scatt} can be enhanced by increasing the field confinement (reducing N), although it is not clear if this mechanism is more convenient to measure Raman signals than conventional cavity-enhanced Raman spectroscopy [100]. Instead, resonant scattering from a high-frequency polariton mode at finite k_{\parallel} can assist in the energy transfer into a lower frequency polariton at lower values of k_{\parallel} for which radiative pumping is inefficient due to little overlap with the bare emission spectrum (see Fig. 1). In conclusion, there is no reason to believe that VAS cannot be a more efficient energy transfer mechanism than radiative pumping to pump low frequency polaritons, despite its $1/N^2$ dependence. However, whether this is the case for the parameter regimes in experiments requires further analysis that will be the focus of future works.”

III. REVIEWER 3

In the context of exciton polaritons that emerge from the strong interaction between molecular excitations and confined photons, the authors derive from a microscopic Holstein-Tavis-Cummings Hamiltonian the two main relaxation mechanisms that have been proposed for these hybrid light-matter quasi-particles: Radiative Pumping (RP) and Vibrationally Assisted Scattering (VAS). Relying on a well established approach to map this Hamiltonian

for identical permutationally symmetric few-level systems into boson operators, they derive expressions for the rates of both mechanisms (RP and VAS) employing Fermi's Golden Rule. Results for numerical simulations illustrate how this approach can be employed to investigate molecular polaritons in the large N limit, typical of experimental realisations of exciton polariton condensation and lasing that build upon polariton relaxation.

This work builds up on previous theoretical developments (references 32,33,60,61,63,64,81) aimed at investigating the relaxation of polaritons in disorder materials with the key difference that it separates RP and VAS by respectively treating photonic and vibronic couplings of the Hamiltonian as first order perturbations. With this strategy, Juan B. Pérez-Sanchez and Joel Yuen-Zhou provide a rationale for recent atomistic molecular dynamics simulations that suggested that the same molecular vibrations drive RP and VAS (reference 77). The main finding of the work is however that, for materials with strong electron-vibron coupling, VAS includes additional terms that arise from higher order terms on the light-matter coupling g . The results are important for both theoreticians and experimentalists in the polariton community and I believe that a derivation from first principles of the rates of both RP and VAS belongs to a journal such as Nature Communications. However, I have a few suggestions for the authors prior publication.

We appreciate the reviewer for the accurate description of our work and their positive comments.

1) The manuscript is well written and the literature related to this work is properly cited. Nevertheless, a few citations are still missing, in particular to guide a general readership less familiar with polaritons. Therefore, I suggest the authors to add work of their choice referring to single molecule strong coupling in page 2, line 26, collective strong coupling in page 2, line 27, the large N problem in page 3, line 50 (besides the work in preparation they make reference to) and the bosonization approach in page 3, line 58 (the authors can here already cite references 47 to 51 so that it is clear that this is not the novelty of the presented work).

We thank the reviewer for pointing out some missing citations that we also consider appropriate. We also re-write the sentence in the introduction where we mention the bosonization approach.

We have included some references to include works on single-molecule and collective strong light-matter coupling. In the manuscript we added: "While single molecules can strongly

couple to confined fields of plasmonic nanocavities [12–16], a more common scenario requires an ensemble of matter excitations collectively coupled to optical modes in microcavities, leading to the emergence of polariton states and a dense manifold of so-called dark states [1–11]”

Regarding the discussion about the large N problem, we have made modifications to that paragraph and included additional citations. In page 2 we added: “First-principle Hamiltonians that go beyond the Holstein-Tavis-Cummings (HTC) model have been put forward with the aim of understanding polariton modified chemical reactivity [40,41], and recent theoretical works suggests that relaxation from polaritons to dark states in the $N \rightarrow \infty$ limit can be understood simply as an optical filtering effect: polaritons act as windows through which vibronic states can be optically excited [42,43]. This is consistent with several theoretical [44,46] and experimental [47,50] works.”

We have removed the sentence saying “The converse, dark state to polariton relaxation processes vanish when $N \rightarrow \infty$ (K. Schwennicke et al., preparation)” from the introduction because it is a result from this work.”

To clarify that the bosonic formalism used in this work has been introduced previously, the sentence where the bosonic formalism is mentioned now reads: “The aim of this work is twofold: first, we re-derive an exact bosonic picture of organic molecular polaritons from first principles to study molecular dynamics under collective light-matter coupling for arbitrary number of molecules N , excitations N_{exc} , and internal vibrational structure of the molecules [51-57]. This mapping is analogous to the Schwinger boson representation of spins [58], and is ideal for numerical simulations using Meyer-Miller mappings [59-62], quantum cumulant expansions [63], quantum computing with bosonic devices [64], and other quantum mechanical methods suited for bosonic systems [65,66].”

2) Separating RP and VAS by respectively treating photonic and vibronic couplings of the Holstein-Tavis-Cummings Hamiltonian as first order perturbations is a well justified approach since the community has considered these mechanisms as radiative and non-radiative relaxation channels, respectively. While the authors admit that the choice of m' for partitioning the Hamiltonian is arbitrary, it would be highly desirable that they provide further guidelines on how to chose it with respect to the system parameters, in particular the light-matter and vibronic couplings.

The problem of partitioning the vibronic coupling into weak and strong components is closely

related to the problem of partitioning a system into a subsystem and a bath; it is somewhat arbitrary. In this case, the separation must be such that it allows for the use techniques like first-order perturbation theory and for the definition of vibrational relaxation rates. Although there are many ways to partition the Hamiltonian to meet this requirement, we must also obtain a partitioning that is physically meaningful. For example, in the problem of triplet harvesting in the collective strong coupling regime, it would be ideal to use all singlet states in the interval $i = 1, m'$ and all triplet states $i = m', m$; the weak vibronic coupling Hamiltonian \hat{H}_{vc} would contain the spin orbit coupling that allows for reverse intersystem crossing.

We realized we may have not explained this very clearly in the original manuscript. We have modified the paragraph in page 7 to explain this better. We write: “This partitioning is based on an important observation: at zero-temperature, all ground-state molecules are in the \hat{b}_1 mode, and light-matter coupling is collectively enhanced at the FC configuration via bosonic stimulation. Therefore, the term $g(\hat{B}_1^\dagger \hat{b}_1 \hat{a} + \hat{B}_1 \hat{b}_1^\dagger \hat{a}^\dagger)$ must be included in \hat{H}_0 , while single-molecule light-matter coupling terms ($\langle \hat{H}_{sm} \rangle \sim g$) can be considered perturbatively (see Appendix B for details). On the other hand, the separation of the vibronic coupling terms into strong and weak, given by m' , is to some degree arbitrary, and resembles the separation of a total Hamiltonian into a system and a bath in open quantum systems. In general, vibronic coupling terms that can lead to vibronic features in the linear response must be included in \hat{H}_0 , while weak vibronic couplings away from the FC region, that are small enough to afford a perturbative treatment, should be included in \hat{H}_{vc} ($\langle \hat{H}_{vc} \rangle \sim W$). Moreover, the definition of \hat{H}_{vc} should be made such that it allows for the definition of physical meaningful relaxation rates. For example, in the problem of triplet harvesting in the collective strong coupling regime [70], defining the weak vibronic coupling Hamiltonian \hat{H}_{vc} as the spin-orbit coupling (singlets are in the interval $i = 1 - m'$ while triplets are in the interval $i = m' - m$) allows for the definition of RISC rates from dark to polaritons states. In quantum optics models where the absorption spectrum showcases only two clear polariton peaks [71], considering all vibronic couplings as weak ($m' = 1$) leads to the definition of vibrational relaxation rates between polaritons and dark states [37,71-73]. More formally, the partitioning can be carried out using effective-mode theory [74-76], chain mappings [77,78], variational polaron transformation [70], among other techniques [79].”

Nonetheless, we would like to highlight that one of our conclusions in this work is that the

framework of vibrational relaxation (and hence the partitioning of the Hamiltonian into a polaritonic Hamiltonian and a vibronic perturbation) is highly inconvenient and confusing, and advise the community to stick to the radiative pumping picture granted that all orders of perturbation theory in g are considered.

In page 16 we added: “Since this approach leads to a much more intuitive understanding of the mechanisms involve in the relaxation between dark and polaritons states, as well as much simpler mathematical expressions for the corresponding rates, we advise the community to stick to the radiative pumping framework”

3) In connection to the previous comment, the authors claim that their work generalizes the formalism proposed by Herrera and Spano (reference 82,83). In order to better show the connection to that work, could the authors identify the same three light-matter coupling regimes described in reference 60 for a material of their choice and comment on whether RP or VAS will constitute the main relaxation mechanism?

We thank the reviewer for asking us this question, as we consider it important to explain our work in reference to previous works that are important and understood in the field. We have read the manuscript by Herrera and Spano carefully and concluded that the three regimes listed in such paper can be explained by radiative pumping. Here is why:

In the paper by Herrera and Spano, the authors calculate one- and two-particle polariton and dark states. Without any coupling between the two, the two-particle polaritons become inaccessible. However, the key aspect of the work is to realize that there is a small coupling that hybridizes one- and two-particle states. Importantly, with our formalism we have identified such coupling as the single-molecule light-matter coupling. The authors of such paper then calculate eigenstates of the total Hamiltonian including such coupling and define the polariton photoluminescence as the result of the cavity leakage from the mixed (one- and two-particle) states. More specifically, the photoluminescence is calculated via the matrix element between one of the hybrid states (we think the one with more one-particle character) and the final state $|g_m \nu_m > 1, 1_c\rangle$ (one molecule vibrationally-excited and the emitted photon, a two-particle state). This gives rise to photoluminescence and frequencies lower to that of the polariton state because of the release of the photon, which is consistent with radiative pumping.

Indeed, we agree that such mechanism is almost identical to the radiative pumping as defined in our formalism, except for the fact that we treat the coupling between one-particle and

two-particle states as a perturbation and define a Fermi’s Golden Rule rate from a one-particle dark state to all the two-particle polariton states. This approach automatically gives you photoluminescence because the cavity leakage is so large compared to single-molecule coupling that the definition of polaritons already include hybridization with the molecular and electromagnetic baths. In other words, the main difference between the two works is the following: the approach followed by Herrera and Spano consists of first hybridizing one- and two-particle states, and then calculating the leakage of the resulting states. On the other hand, our approach consists of first hybridizing one- and two-polariton (and dark) states with their environments, and subsequently calculate rates between them.

The three different regimes encountered by Herrera and Spano in their manuscript can be understood as due to the complex dependence of the bare molecular emission and the polariton transmission spectra on the vibronic coupling and the collective light-matter coupling strengths.

Finally, in order to obtain polariton-assisted Raman scattering (VAS) rates using the approach used by Herrera and Spano, they must also include three-particle polariton states. More specifically, they must hybridize one-, two-, and three-particle polaritons, and calculate the cavity leakage of the resulting eigenstate with the highest one-particle polariton character by projecting it onto the three-particle state $|g_m\nu_m > 1, g_n\nu_n > 1, 1c\rangle$. Since the mixing from one- to three-particle states is proportional to the squared of the coupling between particle “manifolds” (which is proportional to g), this will recover the g^4 ($1/N^2$) dependence of the VAS rate.

We have modified the paragraph in page 6 and 7 where we mentioned the works by Herrera and Spano. It now reads: “This approach shares deep connections with previous works by Herrera and Spano [82,83]. Their work considers only one-particle ($|1\rangle$ and $|e_k\rangle$) and two-particle ($|g_k1\rangle$ and $|g_k e_{k'}\rangle$) states, that are either permutationally or non-permutationally symmetric. Our formalism extends this picture by going beyond two-particle states. Although the one- and two-particle states are enough to account for the mechanism of radiative pumping in different regimes of vibronic coupling and collective light-matter coupling strengths [82], accounting for three-particle states and beyond is important for a complete characterization of polariton photoluminescence. For example, here we show that inclusion of three-particle states ($|g_k g_{k'} 1\rangle$ and $|g_k g_{k'} e_{k''}\rangle$) allows for the characterization of a new mechanism we call polariton-assisted Raman scattering, which we believe corresponds to

the VAS mechanism experimentally observed by Coles et al. [67]. Although we leave out non-permutationally symmetric states in this work, we argue that they cannot be populated if the initial state is permutationally symmetric.”

4) In page 7, line 150, the authors provide an equation to define the first excitation manifold. While for a familiar readership such manifold is obvious, in my opinion the provided equation brings confusion because of the sum over the m vibrational modes. I would suggest a definition making explicit the fact that at most there is one excitation in the system: either there is a photon and all molecules are in their ground state, either one and only one molecule is in its excited state and all other are in the ground state. Related to this: I understand the maximum number of allowed vibrations is $N \times m$. Is that correct? We thank the reviewer for making us aware that the way we introduced the first excitation manifold was neither clear nor properly justified.

Regarding the sum over m in the definition of the number of excitations (the vibrational states) comes about because the sum is counting the number of molecules in all vibrational states of the electronic excited states. The result of that sum is the total number of molecules in the electronic excited state. We agree with the reviewer that this is a bit confusing at first. We instead explain it with simple words and provide a justification to restrict our analysis to the first excitation manifold.

In page 7 we write: “we restrict ourselves to the first excitation manifold (the system only has one electronically-excited molecule or one photon, but arbitrary number of molecules with phonons). This is a good approximation for the linear regime on the interaction between the cavity and the external laser [80].”

Each molecule has m vibrational states (leaving the electronic degree of freedom aside). However, this does not mean that the total number of allowed vibrations would be $m \times N$. In the more general case where molecules are not indistinguishable, the number of allowed vibrational states would be m^N . However, this is not the case here because molecules behave like bosonic particles, and we need to count the total number of configurations in which N balls (molecules) can be distributed amongst m boxes (vibrational states), with no restrictions on how many of them can be in the same box. For example, let us ignore electronic excitations and consider the case $m = 2$ and $N = 3$. The system only has 4 possible vibrational configurations, (3,0), (2,1), (1,2), and (0,3), but $m \times N = 6$.

5) Concerning the polariton-assisted Raman scattering (section V, pages 15-16), does the

model presented by the authors suggest that polaritons could lead to amplified Raman scattering signals?

We thank the reviewer for the very important question.

First, we would like to clarify that the polariton-assisted Raman scattering (or VAS) mechanism that we characterize here is different from the Raman scattering that is measured in Raman spectroscopy. In the latter the light comes from an external source and has a $1/N$ dependence, while in the former the incident light comes from incoherent excitons (dark states) and has a $1/N^2$ dependence (the extra $1/N$ comes from the need for dark state emission). The author is correct in pointing out that these processes are enhanced when increasing the field confinement (via reduction in N). However, if the objective is to obtain amplified Raman signals, VAS is probably not convenient compared to other experiments such as cavity-enhanced Raman scattering. The importance of VAS lies in its potential to provide an efficient channel to pump polaritons at low frequency, where the overlap with the bare molecular emission spectrum makes radiative pumping inefficient.

In page 18-19 we now write: We expect Γ_{scatt} to increase for multimode cavities since dark state (virtual) emission can be resonantly and off-resonantly scattered from the entire lower polariton branch (the sum over all intermediate states $|\xi, \{n_j\}\rangle$ and final states $|\xi', \{n'_j\}\rangle$ in equation (19) must also include a sum over all $k_{||}$ and $k'_{||}$ modes, respectively). Moreover, Γ_{scatt} can be enhanced by increasing the field confinement (reducing N), although it is not clear if this mechanism is more convenient to measure Raman signals than conventional cavity-enhanced Raman spectroscopy [100]. Instead, resonant scattering from a high-frequency polariton mode at finite $k_{||}$ can assist in the energy transfer into a lower frequency polariton at lower values of $k_{||}$ for which radiative pumping is inefficient due to little overlap with the bare emission spectrum (see Fig. 1). In conclusion, there is no reason to believe that VAS cannot be a more efficient energy transfer mechanism than radiative pumping to pump low frequency polaritons, despite its $1/N^2$ dependence. However, whether this is the case for the parameter regimes in experiments requires further analysis that will be the focus of future works.

5) **Technical suggestions:**

5a) The paper could be further improved by relating more strongly the figures to the text. Along the same line, in my opinion, the right panel of Figure 1 does not illustrate in a clear manner the relaxation mechanisms reported in this work.

We agree with the reviewer on this matter.

Since we have provided arguments to justify the expected behavior of the relaxation mechanisms in the multimode case, we have kept only the left panel of Figure 1.

We realized that our lack of reference to the figures is partially due to their lack of clarity. In particular, we significantly improved Figures 4 and 5 to make the mechanisms of radiative pumping, polariton-assisted photon recycling, and polariton-assisted Raman scattering more intuitive to the readers. We have also added references to them in the text when appropriate.

5b) Reference 37 is pointing to the arXiv version of that work while that work has already been accepted in a peer reviewed journal i.e. Dutta et al., Nat Commun 15 (2024) 6600.

We thank the reviewer for the observation.

We updated this and other references that have been updated during the review process.

5c) The name of the first author of reference 59 is incorrect. It is not Martínez but Sánchez Martínez.

We have fixed these mistakes.

5d) Page 13, line 264: there is a typo in the name of the first author of references 31, 64 and 65.

We have fixed these mistakes.

IV. ADDITIONAL MODIFICATIONS TO THE MANUSCRIPT

a) We have slightly changed our notation for the vibronic polariton and dark states as well as their coefficients, so that it is easier for the reader to follow the derivations without the need to look at the SI. Below Eq. 9 (page 9) we added: "... where we renamed $a_{(N-1)\dots 1j\dots}^{(\xi)} \equiv a_{1j}^{(\xi)}$, set $\omega_{g,1} = 0$, and approximated $\omega_{\xi,(N-1)\dots 1j\dots} \approx \omega_{\xi} + \omega_{g,j}$, with $\omega_{\xi} = \omega_{\xi,(N-1)\dots 0\dots}$ being the polariton frequency and ..."

b) We have fixed a typo in Eq. 13 (previously Eq. 12), which now reads: "

$$\hat{H}_m = \sum_{i=1}^2 \omega_{\nu,i} \hat{\beta}_i^\dagger \hat{\beta}_i + \left[\omega_0 + \sum_{i=1}^2 \omega_{\nu,i} \sqrt{s_i} \left(\hat{\beta}_i^\dagger + \hat{\beta}_i \right) \right] |e\rangle \langle e|$$

"

c) We now clarify that the linear optical properties $\text{Im}[D^{(R)}(\omega)]$, $A(\omega)$, and $T(\omega)$, in the formula for radiative pumping (Eq. 11), correspond to $N - 1$ molecules in the cavity instead of N . This difference is negligible in the large N limit. In pages 10-11 we write: “More rigorously, $D^{(R)}(\omega)$, $A(\omega)$, and $T(\omega)$ are the linear optical quantities where one of the molecules is vibrationally-excited while the remaining $N - 1$ are in the global ground state (molecules emit into polaritons with a slightly reduced Rabi splitting $g\sqrt{N-1}$, an effect negligible in the large N limit)”

d) We have included a sentence explaining the physical meaning of the photon-photon correlation function $\text{Im}[D^R(\omega)]$. In page 11 we write: $\text{Im}[D^R(\omega)]$ measures the dissipation rate of the emitted photon into the molecular and electromagnetic baths.

e) We updated Ref. 92 from the arXiv to the published version.

f) In the SI section 3a we included how linear optical properties can be obtained from the Hamiltonian $\hat{H}_{rp}^{(0)}$ and made some modifications to clarify the notation. We added: First, we write the photon density of states in terms of the photon-photon correlation function $D^{(R)}(\omega)$ in the $N \rightarrow \infty$ limit,

$$\begin{aligned} D^{(R)}(\omega) &= \langle N00 \dots 0, 00 \dots 0, 1 | \frac{1}{\omega - \hat{H}_{rp}^{(0)}} | N00 \dots 0, 00 \dots 0, 1 \rangle \\ &= \sum_{\xi} |a_0^{(\xi)}|^2 \frac{1}{\omega - \omega_{\xi} + i\gamma_{\xi}} \\ &\approx \sum_{\xi} |a_{1_j}^{(\xi)}|^2 \frac{1}{\omega - \omega_{\xi} + i\gamma_{\xi}} \\ \text{Im}[D^{(R)}(\omega)] &\approx - \sum_{\xi} |a_{1_j}^{(\xi)}|^2 \frac{\gamma_{\xi}}{(\omega - \omega_{\xi})^2 + \gamma_{\xi}^2}. \end{aligned}$$

Here we added a broadening γ_{ξ} to the eigenvalues of $\hat{H}_{rp}^{(0)}$ due to finite cavity lifetime κ . We also simplified the notation for the photonic Hopfield coefficient as $a_{N \dots 0 \dots}^{(\xi)} = a_0^{(\xi)}$ and $a_{(N-1) \dots 1_j \dots}^{(\xi)} = a_{1_j}^{(\xi)}$, and set $\omega_{g,1} = 0$. The third line comes about from approximating $g\sqrt{N-1} \approx g\sqrt{N}$ (see Appendix I for details). This approximation is valid for large number of molecules where the contraction of the Rabi splitting is negligible compared to the broadening of the spectra.

g) In the SI section 5 we explicitly mention that we derive the vibrational relaxation rate in the linear-vibronic coupling limit, and show the 3×3 matrix that needs to be diagonalized to find the polariton and dark states. **We added:**

The Hamiltonian $\hat{H}_{vr}^{(0)}$ in the weak vibronic coupling limit can be written as

$$\hat{H}_{vr}^{(0)} = \omega_c \hat{a}^\dagger \hat{a} + \sum_i^m \omega_{g,i} \hat{b}_i^\dagger \hat{b}_i + \sum_i^m \omega_{e,i} \hat{B}_i^\dagger \hat{B}_i + g \sum_i^m \left(\hat{B}_i^\dagger \hat{b}_i \hat{a} + \hat{B}_i \hat{b}_i^\dagger \hat{a}^\dagger \right).$$

All vibronic couplings are absent from $\hat{H}_{vr}^{(0)}$ since they are considered perturbations. For simplicity we restrict ourselves to the case where $\langle \varphi_i^{(g)} | \hat{V}_e | \varphi_i^{(g)} \rangle = \langle \varphi_i^{(g)} | \hat{V}_g | \varphi_i^{(g)} \rangle + \omega_0$, e.g., the linear-vibronic coupling model [71].

The Hamiltonian $\hat{H}_{vr}^{(0)}$ commutes with the operators $\hat{n}_i + \hat{n}'_i = \hat{b}_i^\dagger \hat{b}_i + \hat{B}_i^\dagger \hat{B}_i$, which represent the number of molecules on each vibrational excited state i (regardless of their electronic state). We focus on the first excitation manifold, where $\sum_i n'_i = 1$, $n'_i \in \{0, 1\}$. We can write the Hamiltonian $\hat{H}_{vp}^{(0)}$ projected on particular set of quantum numbers $\{n_i + n'_i\}$ as

$$\mathbf{H}_{\text{vc}, \{n_i + n'_i\}}^{(0)} = \begin{pmatrix} \sum_i^m n_i \omega_{g,i} + \omega_c & g\sqrt{n_1} & g\sqrt{n_2} & \cdots & g\sqrt{n_m} \\ g\sqrt{n_1} & \sum_i^m n_i \omega_{g,i} + \omega_0 & 0 & \cdots & 0 \\ g\sqrt{n_2} & 0 & \sum_i^m n_i \omega_{g,i} + \omega_0 & \cdots & 0 \\ \vdots & \vdots & \vdots & \ddots & \vdots \\ g\sqrt{n_m} & 0 & 0 & \cdots & \sum_i^m n_i \omega_{g,i} + \omega_0 \end{pmatrix},$$

The eigenstates of $\hat{H}_{vc}^{(0)}$ can be separated exactly into polaritons and dark states. The polariton states can be written as...

... This dark state is obtained from diagonalizing the Hamiltonian

$$\mathbf{H}_{\text{vc}, (N-1) \dots 1_{\mathbf{k}} \dots}^{(0)} = \begin{pmatrix} \sum_i^m n_i \omega_{g,i} + \omega_c & g\sqrt{N-1} & g \\ g\sqrt{N-1} & \sum_i^m n_i \omega_{g,i} + \omega_0 & 0 \\ g & 0 & \sum_i^m n_i \omega_{g,i} + \omega_0 \end{pmatrix},$$

which in the particle notation yields

$$|D_k\rangle = \sqrt{\frac{N-1}{N}} |e_k\rangle - \frac{1}{\sqrt{N}} |g_k e_1\rangle.$$

h) We have clarified how the bosonic mapping is made. In page we write “We carry out this mapping by transforming the molecular operators $\hat{O} = \sum_i^N \hat{o}_i$ according to the standard

recipe,

$$\hat{O} \rightarrow \sum_{i,j} \langle i|\hat{o}|j\rangle \hat{\mathcal{B}}_i^\dagger \hat{\mathcal{B}}_j,$$

where \hat{o} is a single-molecule operator, and $\hat{\mathcal{B}}_i$ are bosonic operators that annihilate a molecule (not an exciton) in a *vibronic* state $|i\rangle$.

i) Additional changes that do not alter the meaning of the original sentence were made throughout for better readability (improved wording) and to comply with the journal's formatting guidelines (removal of section headers). These do not appear highlighted in red in the main manuscript.

**Radiative pumping vs vibrational relaxation of molecular
polaritons: a bosonic mapping approach**

Answer to reviewers 2

Juan B. Pérez-Sánchez and Joel Yuen-Zhou*

Department of Chemistry, University of California San Diego, La Jolla, CA 92093, USA

(Dated: January 1, 2025)

In black: questions and comments by reviewers.

In blue: answer to reviewers' comments.

In red: modifications of the manuscript addressing reviewer's comments. They also appear highlighted in red in the manuscript file.

I. REVIEWER 1

The authors have clarified most of my questions. However, I remain somewhat skeptical about the publication of this draft in its present form in Nature Communications for the following reasons:

1. It is not at all clear, to what extent the single-mode approximation remains valid, especially for calculating such a relaxation rate given the absence of Frohlich-like scattering. The authors state, "our results can be extended to the multimode scenario owing to a crucial observation by Engelhardt and Cao [90]: the coupling between different cavity k_{\parallel} modes vanishes in the limit where $N \rightarrow \infty$ ". This statement is wrong/misrepresentation of what has been discussed in Ref. 90. In the work of Engelhardt and Cao [90], a multimode disordered Tavis-Cummings model (which does not include phonons) has been considered, where the Heisenberg equation of motion of the cavity modes [see SI of Ref. 90] with different k 's are coupled due to their coupling to the matter degrees of freedom whose onsite energies have Gaussian disorder. It is this coupling that vanishes in the thermodynamic limit. This has nothing to do with the phonon-induced scattering between polariton states (of the same branch), as phonon-induced (dynamic) disorder, even in the semi-classical picture, cannot be simply modeled with static disorder. Therefore, is it incorrect to state that phonon-induced Frohlich-like scattering between polariton states at different k 's is unimportant because a work finds it so when ignoring phonons.

We agree with the reviewer that we do not include Frohlich-like scattering in our model, which is needed for a more complete description of polariton photoluminescence. We have tuned out some of our statements to clarify that Frohlich-like scattering processes are not accounted for in our model and care must be taken when extrapolating our results to the multimode scenario.

* joelyuen@ucsd.edu

We replaced the sentence from page 9 lines 167-169 that read “accounting for three-particle states and beyond is important for a complete characterization of polariton photoluminescence.” We now write “accounting for three-particle states and beyond is required in the characterization of polariton photoluminescence.”

We have modified the last paragraph of page 11. It now reads: “A simple extension of equation (11) to the multimode scenario can be justified owing to a crucial observation. As Engelhardt and Cao recently showed, in the absence of vibronic coupling, coupling between different cavity $k_{||}$ modes vanishes in the limit where $N \rightarrow \infty$ [93]. If we assume this to hold in the presence of homogeneous broadening, this implies that $k_{||}$ is a good quantum number for the polaritonic eigenstates of $\hat{H}_{rp}^{(0)}$, which is the total Hamiltonian in the $N \rightarrow \infty$ limit [44]. This allows us to define linear optical properties $D_{k_{||}}^{(R)}(\omega)$, $A_{k_{||}}(\omega)$, and $T_{k_{||}}(\omega)$. For finite N , the single-molecule coupling \hat{H}_{sm} that gives rise to radiative pumping couples different cavity modes. Therefore, we can estimate the radiative pumping rate in the multimode case to take the form

$$\Gamma_{rp} = \frac{2g^2}{\kappa} \sum_{k_{||}} \int d\omega \sigma_{em}(\omega) \left[A_{k_{||}}(\omega) + 2T_{k_{||}}(\omega) \right], \quad (1)$$

where we sum over all the $k_{||}$ modes. This expression is consistent with the fact that radiative pumping is simply fluorescence from incoherent excitons, and can pump any polariton mode. **However, this argument neglects Fröhlich interactions that could result from collective phonons. This limitation arises from the starting point of our model whereby no intermolecular couplings are present in the absence of the photon mode; these interactions might be more relevant in structured solids [94].**”

2. I therefore fully disagree with the author’s decision to keep Fig.1, in its present form that illustrates a polaritonic dispersion that is fully absent from the theoretical model considered here. In fact, it is because the polariton community is familiar with the polariton dispersion, this figure will cause confusion, as seeing this would lead one to expect the work and the analytical expression derived to have considered the Fröhlich-like scattering processes. I am sorry, but I cannot recommend this work unless this figure is made accurate.

The goal of Fig. 1 was to show the photophysical processes characterized in this work (radiative pumping, polariton-assisted photon recycling, and polariton-assisted Raman scattering), using an illustration familiar to the polaritonics community. Since our model does not include Frohlich-like scattering, it is absent from Fig. 1. We agree with the reviewer that this is misleading as the reader may think that the absence of other mechanisms in Fig. 1 implies that no other scattering mechanisms exists. A few other statements in the manuscript also wrongly suggests this to be the case. Therefore, we have removed Fig. 1 and replaced it with a figure where the polariton dispersion is not shown.

We have removed Fig. 1 and replaced it with the following figure:

FIG. 1. Schematic representation of the polariton relaxation mechanisms categorized in this work. a) Radiative pumping: a first-order processes in the single-molecule light-matter coupling. An emitted photon from an incoherent exciton can either leak out of the cavity or be reabsorbed by the molecules, enabling polariton-assisted photon recycling. b) Polariton-assisted Raman scattering: a second-order process in the single-molecule light-matter coupling. It involves the virtual emission of a photon from an incoherent exciton, followed by Raman scattering by a second molecule. The resulting red-shifted photon then leaks out of the cavity. Vibrational relaxation encompasses radiative pumping, polariton-assisted Raman scattering, and higher-order processes.

We thank the reviewer for their constructive feedback. In particular for their comments on the limitations of our single-mode approximation and the overlook of Frohlich-like scattering processes.

II. REVIEWER 2

Pérez-Sánchez and Yuen-Zhou have thoroughly addressed my previous questions and comments, particularly regarding the various contributions in a complex polariton system involving multimode interactions. The revised manuscript further clarifies the independence between vibronic coupling and light-matter interactions in the finite-N molecule regime, as well as the role of cavity photons in the strongly coupled photon-cycling scheme. Additionally, the authors have expanded the theoretical description to encompass a broader range of k-values, potentially linking it to the Bose scattering mechanism. Overall, the authors' responses are satisfactory, and I recommend publication in Nature Communications.

We thank the reviewer for the constructive feedback and previous comments, which led to an improvement of the manuscript.

III. REVIEWER 3

The authors have satisfactorily addressed most of my comments and the manuscript has significantly gained in both clarity and position within the field despite that the Figures remain poorly related to the text. I have two additional minor comments prior publication:
Comment 1: With the increased clarity of the work and the lengthier description of the so-called “photon-recycling” mechanism, I now understand that this corresponds to population exchanges between dark and bright states mediated by photons that get emitted and re-absorbed in the polariton system. Since the authors additionally stress that this mechanism is responsible for long range polariton transport, I believe work dedicated exclusively to the study of polariton propagation and showing the importance of this same mechanism, although differently labelled, should be cited i.e. Sokolovskii et al., Nat. Commun. 14 (2023) 6613 and Balasubrahmaniyam et al. Nat. Materials 22 (2023) 338.

We thank the reviewer for the constructive feedback and for pointing out the missing references.

We have included the two references. They are Refs. [20] (Sokolovskii et al.) and [28] (Balasubrahmaniyam et al.).

Comment 2: Although in the original version of the manuscript, the authors had already drawn attention on some of the possible limitations of their model because of the single cavity mode approximation, comment 5 of Reviewer 1 is of outmost importance. If at present it is no possible to account for the entire cavity description, the authors should at least tone down their message by avoiding strong statements as the one in page 8, lines 167-169 “accounting for three-particle states and beyond is important for a complete characterization of polariton photoluminescence” when polariton photoluminescence, specially in the low excitation limit, cannot actually be “completely characterized” because of the single mode approximation.

We agree with the reviewer and we have modified that and other statements in manuscript to clarify that there are additional phenomena that our model cannot account for due to our limited description of the cavity modes.

We made the following modifications:

We replaced the sentence from page 9 lines 167-169 that read “accounting for three-particle states and beyond is important for a complete characterization of polariton photoluminescence.” We now write “accounting for three-particle states and beyond is required in the characterization of polariton photoluminescence.”

We have modified the last paragraph of page 11. It now reads: “A simple extension of equation (11) to the multimode scenario can be justified owing to a crucial observation. As Engelhardt and Cao recently showed, in the absence of vibronic coupling, coupling between different cavity $k_{||}$ modes vanishes in the limit where $N \rightarrow \infty$ [93]. If we assume this to hold in the presence of homogeneous broadening, this implies that $k_{||}$ is a good quantum number for the polaritonic eigenstates of $\hat{H}_{rp}^{(0)}$, which is the total Hamiltonian in the $N \rightarrow \infty$ limit [44]. This allows us to define linear optical properties $D_{k_{||}}^{(R)}(\omega)$, $A_{k_{||}}(\omega)$, and $T_{k_{||}}(\omega)$. For finite N , the single-molecule coupling \hat{H}_{sm} that gives rise to radiative pumping couples different

cavity modes. Therefore, we can estimate the radiative pumping rate in the multimode case to take the form

$$\Gamma_{rp} = \frac{2g^2}{\kappa} \sum_{k_{\parallel}} \int d\omega \sigma_{em}(\omega) \left[A_{k_{\parallel}}(\omega) + 2T_{k_{\parallel}}(\omega) \right], \quad (2)$$

where we sum over all the k_{\parallel} modes. This expression is consistent with the fact that radiative pumping is simply fluorescence from incoherent excitons, and can pump any polariton mode. **However, this argument neglects Fröhlich interactions that could result from collective phonons. This limitation arises from the starting point of our model whereby no intermolecular couplings are present in the absence of the photon mode; these interactions might be more relevant in structured solids [94].**”

Also, concerning comment 4 of Reviewer 1, while I understand that Figure 1 is an illustrative figure with which the polariton community is familiar owing in particular to reference 67 [Coles et al. Adv. Funct. Mater. 21 (2011) 3691], the authors should stress in the caption of that figure that the entire polariton dispersion is not accounted for in the present work.

We agree that the reviewer’s suggestion may be a good fix. However, to avoid confusion, we have removed Fig. 1 and replaced it with another figure where the polariton dispersion is not shown.

We have removed Fig. 1 and replaced it with the following figure:

FIG. 2. Schematic representation of the polariton relaxation mechanisms categorized in this work. a) Radiative pumping: a first-order processes in the single-molecule light-matter coupling. An emitted photon from an incoherent exciton can either leak out of the cavity or be reabsorbed by the molecules, enabling polariton-assisted photon recycling. b) Polariton-assisted Raman scattering: a second-order process in the single-molecule light-matter coupling. It involves the virtual emission of a photon from an incoherent exciton, followed by Raman scattering by a second molecule. The resulting red-shifted photon then leaks out of the cavity. Vibrational relaxation encompasses radiative pumping, polariton-assisted Raman scattering, and higher-order processes.